# Toward Real-world Text Image Forgery Localization: Structured and Interpretable Data Synthesis

**Zeqin Yu**[1]  **Haotao Xie**[2]  **Jian Zhang**[3*] **Jiangqun Ni**[1,4*] **Wenkan Su**[3]  **Jiwu Huang**[5]

[1]Sun Yat-sen University  [2]Beihang University  [3]Guangzhou University
[4]Peng Cheng Laboratory  [5]Shenzhen MSU-BIT University

## Abstract

Existing Text Image Forgery Localization (T-IFL) methods often suffer from poor generalization due to the limited scale of real-world datasets and the distribution gap caused by synthetic data that fails to capture the complexity of real-world tampering. To tackle this issue, we propose Fourier Series-based Tampering Synthesis (FSTS), a structured and interpretable framework for synthesizing tampered text images. FSTS first collects 16,750 real-world tampering instances from five representative tampering types, using a structured pipeline that records human-performed editing traces via multi-format logs (e.g., video, PSD, and editing logs). By analyzing these collected parameters and identifying recurring behavioral patterns at both individual and population levels, we formulate a hierarchical modeling framework. Specifically, each individual tampering parameter is represented as a compact combination of basis operation–parameter configurations, while the population-level distribution is constructed by aggregating these behaviors. Since this formulation draws inspiration from the Fourier series, it enables an interpretable approximation using basis functions and their learned weights. By sampling from this modeled distribution, FSTS synthesizes diverse and realistic training data that better reflect real-world forgery traces. Extensive experiments across four evaluation protocols demonstrate that models trained with FSTS data achieve significantly improved generalization on real-world datasets. Dataset is available at Project Page.

## 1 Introduction

In the digital era, text images have become increasingly prevalent in domains such as finance, insurance, and certification audits, serving as essential digital records. As critical credentials containing rich textual and numerical information, they have become prominent targets for forgery. From falsified documents to manipulated news screenshots, such fraudulent alterations pose a serious threat to the authenticity and credibility of digital information.

To address such an issue, researchers have proposed various text image forgery localization (T-IFL) methods [5, 6, 13, 39, 21, 32, 45, 48] that aim to identify the manipulated regions within tampered text images. Early approaches primarily relied on handcrafted features to capture forgery artifacts in text images, such as character misalignment [5, 21], font inconsistencies [6], or layout irregularities [13, 39]. However, with the advancement of tampering techniques, the effectiveness of traditional methods has significantly declined. In response, recent research has increasingly focused on deep learning-based T-IFL methods [32, 45, 48]. Despite their promising performance, the development of efficient and highly generalizable deep learning models often depends on access to large-scale, high-quality datasets containing tampered text images. Unfortunately, creating such datasets remains a significant challenge, as pixel-level manipulation and annotation require time-consuming and labor-intensive efforts by experts. Consequently, the scale of existing real-world

---

*Corresponding authors.

Figure 1: *Visible* vs. *invisible distributions* in synthetic tampered text image datasets. Existing datasets mainly focus on visible attributes (a–d), while our FSTS strategy models invisible tampering parameters (e–g) derived from real-world tampering scenarios.

manually tampered datasets [2, 45, 48, 49] remains relatively limited, hindering the training of generalizable models.

To overcome the data scarcity bottleneck, recent studies [32, 48, 42, 41] have explored synthetic approaches to automatically generate tampered text image datasets. Some approaches, such as DocTamper [32], follow rule-based pipelines that apply predefined tampering types to text images, aiming to generate large-scale datasets. Other methods [41, 42] leverage deep generative models, such as GANs [22] and Diffusion Models [34], to synthesize or modify text regions in scene images, focusing on visual realism. However, these synthetic methods primarily focus on visible attributes in tampered text image datasets, such as scene variety, dataset scale, language diversity, and imaging device differences. These characteristics, largely inherited from general visual analysis tasks (e.g., image classification, segmentation, and object detection), are easily perceptible to human observers, as illustrated in Fig. 1(a–d). We refer to these explicitly visible attributes as the *Visible Distribution*. In contrast, real-world text image tampering often involves complex and invisible combinations of tampering parameters. Forgers tend to select different tampering types [45, 49] (e.g., copy-move, splicing, removal, insertion, replacement) based on the specific scenario and then apply a series of specific main processing (e.g., region selection, text insertion, and other geometric transformations), followed by multiple post-processing operations (e.g., blurring, filtering, color adjustment, JPEG recompression, among others), as shown in Fig. 1(e-g). The resulting high-dimensional tampering parameter vectors extracted from the above tampered images, shaped by multi-step decision-making processes, are visually imperceptible yet critically influence the diversity and subtlety of forgery traces in synthetic samples, thereby limiting model generalization in real-world scenarios. We refer to such complex and invisible combinations of tampering parameters as the *Invisible Distribution*. Therefore, accurately modeling the high-dimensional and concealed Invisible Distribution during data synthesis to generate more representative and diverse training samples and improve model generalization under complex tampering scenarios remains a critical challenge.

To address this challenge, we propose a novel structured and interpretable framework, termed **Fourier Series-based Tampering Synthesis (FSTS)**, which models the Invisible Distribution to simulate complex real-world tampering parameters and enhance the generalization of T-IFL models. Our approach comprises three key steps: (1) we design a structured pipeline to collect tampering parameters from human-performed editing traces, recruiting 67 experts and volunteers to create 16,750 real-world instances across five representative tampering types, averaging 250 samples per participant. Operation histories are automatically recorded through multi-format logs (e.g., video, PSD, and editing logs). (2) We analyze the collected parameters and observe recurring behavioral patterns at both the individual and population levels. Based on this, we formulate a hierarchical distribution modeling framework, where each individual-level distribution is represented as a compact combination of representative basis operation–parameter configurations, and the population-level distribution is constructed by aggregating these individual behaviors. This formulation draws inspiration from Fourier series, enabling a compact and interpretable approximation using basis functions and their learned weights. (3) We then synthesize tampered images by sampling operation–parameter configurations and their frequencies from the modeled distribution, generating diverse and realistic training data that better align with real-world forgery traces. Extensive experiments conducted across four evaluation protocols demonstrate the superior generalization of FSTS-trained models in real-world scenarios.

## 2 Related Work

In recent years, the general image forgery localization (IFL) task has gained widespread attention, with several research teams constructing and releasing publicly available tampered image datasets [2, 10–

12, 23, 20, 30, 24, 45–49] for training and evaluation. Most of these datasets [10, 12, 23, 20, 30, 11, 24] are designed for natural image forgery localization (N-IFL), primarily focusing on natural image scenes such as portrait images, landscape photographs, or general object images. Meanwhile, another ubiquitous form, i.e., text images [2, 45, 48, 49], encountered in documents, invoices, and news screenshots, contain extensive sensitive textual and numerical information, making them highly susceptible to counterfeiting and fraud. However, natural images and text images exhibit significant differences in content structure, semantic density, the spatial distribution of forgery regions, and other aspects. Consequently, models trained on natural-image datasets often struggle to generalize well to T-IFL tasks in real-world scenarios [45, 48].

To tackle the above issue, several studies [2, 45, 48, 49] have developed specialized tampered text image datasets for T-IFL. For example, FindIt [2] constructed a tampered receipt dataset covering essential fields such as amounts, dates, and receipt numbers, consisting of 240 samples. STFD [45] focused on smartphone screenshot images, assembling 4,094 tampered instances derived from genuine text content (e.g., social chat records, e-commerce transactions, and web news screenshots), and modified by dozens of experts using five representative tampering types (e.g., copy-move, splicing, removal, insertion, and replacement) specifically designed for text images. Additionally, CertificatePS [49] released a tampered certificate dataset comprising 4,840 images, captured under indoor and outdoor settings using 77 different mobile phones. The tampered images were created by 25 experts, primarily targeting high-sensitivity areas such as signatures and seals. Moreover, some commercial organizations have organized T-IFL challenges [1, 3, 38], with datasets partly derived from the aforementioned sources [2, 45] or covering similar application scenarios [1]. Although these datasets have significantly improved model adaptability in specific text image tampering scenarios, their construction still heavily relies on expert manual annotation and is constrained by data collection costs, privacy concerns, and the limited scale and diversity of real-world text image scenarios.

To overcome the limitations of current real-world datasets, recent efforts [48, 32, 14, 27, 40–42] turned to the automatic generation of large-scale synthetic tampered text image datasets. PS-scripted [48] proposed a synthetic tampered dataset constructed from book cover images, where random splicing followed by blurring and smoothing was applied to mimic realistic forgery traces. However, the dataset lacks textual and structural semantics and supports only a single tampering type, limiting its representativeness. DocTamper [32] introduced a large-scale synthetic tampered dataset consisting of 170,000 document images. It employs a structured process to separate the foreground and background and applies common tampering types (e.g., copy-move, splicing, and replacement) to the foreground text areas. Despite its scale, this approach relies on predefined, single-rule procedures and fails to capture the diversity of tampering strategies, operation sequences, and post-processing techniques observed in real-world scenarios. Additionally, other works [14, 27, 40–42] have explored deep generative models, such as GANs [22] and Diffusion Models [34], to synthesize or repair localized text regions in scene images. While some of these methods are designed for non-forensic applications like poster design [14, 27, 40], and others involve forgery-related tasks [41, 42] but still emphasize surface-level visual realism over the underlying behaviors and trace diversity of genuine tampering. As a result, existing synthetic datasets often suffer from pronounced distribution gaps compared to real-world tampered text image datasets, primarily due to their failure to adequately model complex tampering strategies and parameter configurations that characterize real-world forgeries. This gap severely limits the generalization ability of current models in practical T-IFL settings and highlights the urgent need for a more principled synthesis framework that captures the invisible distributions underlying real-world tampering.

## 3 The Proposed Synthesis Dataset

### 3.1 Preliminary

**Motivation.** Although existing synthetic datasets have advanced the development of T-IFL by emphasizing observable visual attributes, they often overlook the latent, behavior-level aspects of tampering. These aspects are associated with different tampering types, each comprising specific main processing and post-processing operations parameterized by implementation details, which we term the *invisible distribution*. This gap between synthetic and real-world data in this latent space leads to limited model generalization. To address this, we attempt to explicitly model and simulate such invisible distributions in a structured and interpretable manner.

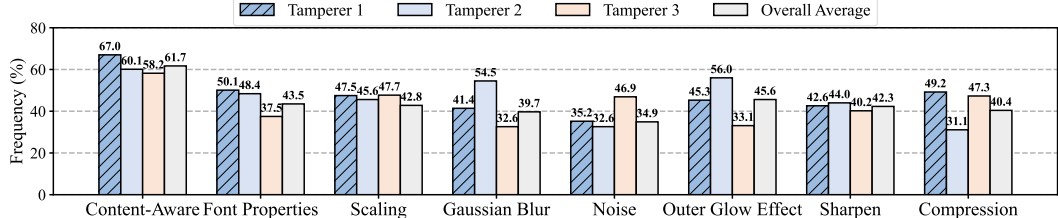

Figure 2: Parameter usage frequencies in "Replacement" tampering samples across three tamperers and the overall average.

**Problem Definition.** Let $t$ denote the tampering parameters, including those associated with the main processing and post-processing operations across different tampering types, as introduced in *Motivation*. We define the real-world tampering distribution as $P_R(t)$, and the synthesized tampering distribution as $P_S(t)$. Our goal is to minimize the discrepancy between $P_S(t)$ and $P_R(t)$ by modeling the underlying $t$:

$$\min D(P_S(t), P_R(t)), \tag{1}$$

where $D(\cdot)$ denotes a distribution distance metric.

**Challenges.** An intuitive approach to the above objective is to collect a sufficient number of $t$ from real-world scenarios, analyze their statistical properties (e.g., distribution patterns and frequencies), and leverage these characteristics to construct a model for $P_S(t)$. However, this approach faces two key challenges in practical applications:

- **How to effectively collect $t$?** Most tampered text images retain only the final output, without operation history, making it difficult to recover the underlying $t$. Therefore, a mechanism is required to infer or extract these $t$ based on the tamperer's operation process.

- **How much $t$ is sufficient to ensure the adequacy of distribution modeling?** Given the diversity and complexity of tampering operations, a small number of parameter samples is unrepresentative and prone to biased modeling. However, exhaustively collecting $t$ to fully characterize $P_R(t)$ is impractical, due to high annotation costs and privacy constraints. A principled strategy is needed to generalize from a limited yet representative subset while preserving the diversity of real-world tampering behaviors.

### 3.2 Our Insights

To address the challenges outlined in Sec. 3.1, we introduce two key insights: one for collecting real-world tampering parameters $t$, and another for modeling their distribution in a structured and interpretable manner.

**Insight 1: Collecting $t$ from Real-World Scenarios.** To address Challenge 1, we design a structured pipeline to collect $t$ from realistic tampering processes. Inspired by previous work [45], we consider five representative tampering types for text image forgery, i.e., copy-move, splicing, removal, insertion, and replacement. We recruit 67 experts and volunteers to perform text image tampering tasks using Photoshop across diverse visible distribution scenarios (e.g., photographic, screenshot, and scanned images). During the editing process, the corresponding tampering parameters $t$ were automatically recorded, resulting in 16,750 tampering instances, with an average of 250 samples per tamperer. The recorded parameters were saved in multiple formats, including video recordings, Photoshop-exported history logs, and project files (.psd). These records provide fine-grained information about the tampering process, capturing operation sequences, parameter values, and layer-level edits, thereby enabling the detailed reconstruction of each tampering instance.

**Insight 2: Hierarchical modeling of $t$.** Based on Insight 1, we address Challenge 2 by conducting a comprehensive analysis of tampered samples across all five tampering types to characterize the distribution of $t$. Among them, we highlight the "Replacement" type as a representative case (Fig. 2), presenting the eight most frequently used parameters aggregated across all tamperers, along with

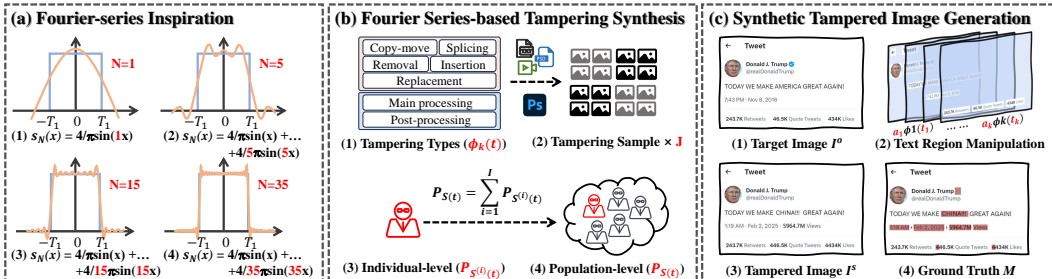

Figure 3: Overview of the proposed FSTS framework. (a) **Inspiration:** A rectangular signal $s(x)$ is approximated by a weighted sum of sinusoidal basis functions, i.e., $s_N(x) = \sum_{k=1}^{N} \frac{4}{(2k-1)\pi} \sin\big((2k-1)x\big)$, where $\sin\big((2k-1)x\big)$ is a basis function and $\sum_{k=1}^{N} \frac{4}{(2k-1)\pi}$ is its weight. Larger $N$ yields higher-fidelity reconstruction, illustrating the idea of decomposition and recombination over a quasi-periodic domain. (b) **Modeling:** Each individual distribution $P_S^{(i)}(t)$ is modeled as a weighted combination of basis tampering configurations (individual-level reconstruction), and their aggregation $P_S(t)$ approximates $P_R(t)$ (population-level reconstruction). (c) **Generation:** Based on the learned basis functions and weights from (b), parameter configurations are sampled and applied to text images, yielding synthetic tampered images that more accurately reflect real-world forgery traces.

the individual configurations from three randomly selected tamperers. From this analysis, two key patterns were observed:

- **Observation 1 – Individual-level recurrence:** Despite differences in visible distribution scenarios, individual tamperers tended to repeatedly adopt similar parameter configurations, reflecting habitual preferences. For example, across all of Tamperer 1's Replacement samples, Content-Aware Fill was used to erase text in 67.0% of the cases, followed by insertion and the application of Gaussian blur (41.4%) and Noise (35.2%) to conceal modifications.

- **Observation 2 – Population-level recurrence:** Certain parameter choices consistently emerged across tamperers. As revealed by the aggregated averages, 61.7% of tamperers applied Content-Aware Fill, while Gaussian blur and Noise addition appeared in 39.7% and 34.9% of samples, respectively, indicating shared tendencies in tampering practices.

These observations reveal that, despite their apparent diversity, tampering behaviors follow recurring patterns at both the individual and population levels. Such structured recurrence suggests that the underlying tampering distribution $P_R(t)$ can be effectively approximated by a compact set of representative operation–parameter configurations and their associated weights. Inspired by this, we introduce a hierarchical modeling framework termed **Fourier Series-based Tampering Synthesis (FSTS)**, which approximates $P_R(t)$ using interpretable basis configurations and their learned weights. As illustrated in Fig. 3(a), just as a waveform can be decomposed into a weighted sum of basis functions, FSTS represents each tampering parameter $t$ as a weighted combination of basis components, where each basis corresponds to a distinct operation–parameter configuration. At the *individual level*, we model each tamperer's behavior as a sparse combination of these basis components. At the *population level*, aggregating individual patterns enables us to yield a compact, interpretable approximation of $P_R(t)$. This formulation offers a principled foundation for synthesizing diverse and realistic tampered samples that more accurately reflect real-world forgery traces.

### 3.3 Fourier Series-based Tampering Synthesis

Building upon the empirical observations in Insight 1 and 2, we now instantiate our proposed FSTS framework (Fig. 3(b)). FSTS hierarchically models $t$ using a set of interpretable basis functions and their associated weights. We describe the individual- and population-level formulations below.

**Individual-Level Tampering Distribution.** The tampering parameter of each individual is modeled as a combination of $K$ predefined tampering types $\phi_k$ ($k = 1, \ldots, K$), each associated with concrete operation–parameter configurations $t_{j,k}^{(i)}$. Here, $t_{j,k}^{(i)}$ denotes the tampering parameter of the $j$-th

instance of type $\phi_k$ performed by individual $i$, where $i = 1, \ldots, I$ and $j = 1, \ldots, J$, and $a_{j,k}^{(i)}$ is the corresponding frequency coefficient. Formally, the individual-level tampering distribution $P_S^{(i)}(t)$ is expressed as:

$$P_S^{(i)}(t) = \sum_{k=1}^{K} \sum_{j=1}^{J} a_{j,k}^{(i)} \phi_k(t_{j,k}^{(i)}). \tag{2}$$

In practice, as revealed in Insight 2 (Observation 1), individual tamperers tend to reuse similar configurations. As the number of samples (i.e., $J$) grows, the configuration statistics for each tampering type converge to stable values, allowing us to approximate each type with a representative setting. Thus, we simplify $P_S^{(i)}(t)$ as:

$$P_S^{(i)}(t) \approx \lim_{J \to \infty} \sum_{k=1}^{K} \left( \sum_{j=1}^{J} a_{j,k}^{(i)} \right) \phi_k(t_{j,k}^{(i)}) = \sum_{k=1}^{K} a_k^{(i)} \phi_k(t_k^{(i)}), \tag{3}$$

where $a_k^{(i)} = \sum_{j=1}^{J} a_{j,k}^{(i)}$ denotes the expected weight of type $\phi_k$ for individual $i$, and $t_k^{(i)}$ is the representative operation–parameter configuration of that type. We select the representative configuration $t_k^{(i)}$ for each tampering type as the instance most frequently observed across all samples of that type, subject to a minimum usage threshold (e.g., $\geq 2\%$), ensuring that only recurrent behaviors are incorporated into our basis set.

**Population-Level Tampering Distribution.** We construct the population-level distribution $P_S(t)$ by aggregating the individual distributions $P_S^{(i)}(t)$ across all tamperers:

$$P_S(t) = \sum_{i=1}^{I} P_S^{(i)}(t) = \sum_{i=1}^{I} \left( \sum_{k=1}^{K} a_k^{(i)} \phi_k(t_k^{(i)}) \right). \tag{4}$$

Given that tamperers often share common configuration preferences (see Insight 2, Observation 2), accurately approximating population-level behavior does not require an excessively large number of participants. Assuming that our collected samples capture sufficient diversity, we simplify the distribution by taking the limit as $I \to \infty$. As $K$ is predefined and finite, we interchange the summation order to aggregate each type's contribution across all individuals before combining them into the final distribution:

$$P_S(t) \approx \lim_{I \to \infty} \sum_{k=1}^{K} \left( \sum_{i=1}^{I} a_k^{(i)} \phi_k(t_k^{(i)}) \right) = \sum_{k=1}^{K} a_k \phi_k(t_k), \tag{5}$$

where $a_k = \sum_{i=1}^{I} a_k^{(i)}$ denotes the aggregated frequency coefficient for tampering type $\phi_k$. Likewise, $t_k$ is selected from $\{t_k^{(i)}\}$ as the configuration shared by at least 5% of the individuals, ensuring that only broadly recurring patterns are retained at the population level.

**Real-World Tampering Distribution.** We define the real-world tampering distribution $P_R(t)$ as a weighted combination of predefined tampering types $\phi_k$, consistent with the modeling basis in Eq. 5. Here, $\hat{t}_k$ denotes the complete set of operation–parameter configurations of type $\phi_k$ assumed to exist in real-world scenarios. Formally, $P_R(t)$ can be written as follows:

$$P_R(t) = \sum_{k=1}^{K} \hat{a}_k \phi_k(\hat{t}_k), \tag{6}$$

where $\hat{a}_k$ denotes the frequency of tampering type $\phi_k$ in real-world data.

**Minimizing Distribution Difference.** As outlined in Eq. 1, our goal is to minimize the discrepancy between $P_S(t)$ and $P_R(t)$. However, directly minimizing the difference between these two complex distributions can be particularly challenging. To simplify the task, we express both as weighted combinations over the same set of basis configurations. Since the synthesized configurations $t_k$ are

Table 1: Overview of the four experimental protocols, summarizing the training and testing settings.

| No. | Protocol | Training | Testing |
|-----|----------|----------|---------|
| 1 | Synthetic → Synthetic | DocT-T, FSTS-T | DocT-S, FSTS-S |
| 2 | Synthetic → Real | DocT-T, FSTS-T | FSTS-1.5k, AFAC, CertificatePS, STFD, FindIt |
| 3 | Real → Real | CertificatePS | FSTS-1.5k, AFAC, CertificatePS, STFD, FindIt |
| 4 | Synthetic Pretraining + Real Fine-Tuning | Pretrained model from Protocol 1 or 2 + Fine-tune on CertificatePS | FSTS-1.5k, AFAC, CertificatePS, STFD, FindIt |

derived from recurring patterns observed in real-world data, we assume $t_k \approx \hat{t}_k$, and simplify the problem by focusing on aligning the coefficients:

$$
\min_{\{a_k, t_k\}} D\left(P_S(t), P_R(t)\right) = \min_{\{a_k, t_k\}} D\left(\sum_{k=1}^{K} a_k \phi_k(t_k), \sum_{k=1}^{K} \hat{a}_k \phi_k(\hat{t}_k)\right)
$$
$$
\implies \min_{\{a_k\}} D\left(\{a_k\}, \{\hat{a}_k\}\right) \quad \text{(assuming } t_k \approx \hat{t}_k\text{)}.
$$
(7)

By reformulating the objective in coefficient space, we sidestep the difficulty of directly matching complex tampering distributions and instead concentrate on aligning the synthesized weights $\{a_k\}$ with their real-world counterparts $\{\hat{a}_k\}$, assuming the basis configurations $t_k \approx \hat{t}_k$. Details of the representative operation–parameter configurations and their empirical frequencies are summarized in the Appendix.

**Synthetic Image Generation.** Once the population-level tampering parameters $\{a_k, t_k\}_{k=1}^{K}$ are obtained, we synthesize tampered images by applying corresponding tampering operations on the original image $I^o$, as illustrated in Fig. 3(c). Formally, the generation process is formulated as:

$$
I^s = \text{Generator}\left(I^o \mid \{a_k, t_k, \phi_k\}_{k=1}^{K}\right),
$$
(8)

where $\text{Generator}(\cdot \mid \cdot)$ denotes the tampering synthesis pipeline (e.g., implemented using Photoshop). Each tampering type $\phi_k$ is executed with its corresponding configuration $t_k$ and synthesized weight $a_k$ to the original image $I^o$. The resulting image $I^s$ embodies tampering patterns consistent with the learned distribution $P_S(t)$, thereby yielding realistic and diverse samples for model training. Implementation details and the full synthesis pipeline are described in the Appendix.

## 4 Experiments

### 4.1 Dataset and Experimental Protocols

To validate the effectiveness of our proposed FSTS strategy, we conduct experiments on both synthetic and real-world datasets. As one of the largest public synthetic datasets for T-IFL, DocTamper [32] serves as our baseline for constructing comparable training and testing protocols. For the synthetic setting, we follow the DocTamper-Train (DocT-T) protocol [32] and sample 50,000 text images to construct the training set. We then apply our proposed FSTS strategy to the same set [8, 17, 26, 37] to generate FSTS-Train (FSTS-T). Similarly, we use dataset [18], consistent with the second cross-domain setting in DocTamper (DocT-S), to construct FSTS-S for cross-domain testing. In addition, we evaluate generalization on five real-world datasets: FSTS-1.5k (a held-out subset of 1,488 real images excluded from parameter modeling in FSTS), AFAC [1], CertificatePS [49], STFD [45], and FindIt [2]. Further details on these datasets are provided in the Appendix. To systematically assess the impact of synthetic data and evaluate model performance under different training and testing settings, we define four evaluation protocols, as summarized in Table 1.

- **Protocol 1: Synthetic Data Training with Synthetic Data Testing.** Following the evaluation protocol of DocTamper [32], we train the models on DocT-T and FSTS-T and evaluate them on DocT-S and FSTS-S. This setting provides a controlled benchmark for training and evaluating models on synthetic tampering patterns.

- **Protocol 2: Synthetic Data Training with Real-World Data Testing.** Using the model trained in Protocol 1, this protocol evaluates its generalization capability by testing on

Table 2: Pixel-level F1 and AUC performance of T-IFL for Protocols 1 and 2, showing models trained on synthetic datasets (DocT-T and FSTS-T) and tested on both synthetic and real-world datasets. Each method includes three rows representing different training–testing configurations. The first and second rows show results for models trained on DocT-T and FSTS-T, respectively. The third row (**Gain** $\Delta$) shows the performance difference between FSTS-T and DocT-T (FSTS-T minus DocT-T). Positive gains are highlighted in red, and negative gains in blue.

| Methods | Test / Train | Synthetic | | | | Real-World | | | | | | | | | | | | |
|---|---|---|---|---|---|---|---|---|---|---|---|---|---|---|---|---|---|
| | | DocT-S | | FSTS-S | | FSTS-1.5k | | AFAC | | CertificatePS | | STFD | | FindIt | | Average | |
| | | F1 | AUC | F1 | AUC | F1 | AUC | F1 | AUC | F1 | AUC | F1 | AUC | F1 | AUC | F1 | AUC |
| RRU-Net [7] | DocT-T | .501 | .968 | .253 | .864 | .215 | .696 | .088 | .798 | .383 | .782 | .099 | .772 | .211 | .776 | .199 | .765 |
| | FSTS-T | .214 | .828 | .401 | .881 | .541 | .933 | .307 | .874 | .433 | .863 | .177 | .855 | .252 | .793 | .342 | .864 |
| | **Gain** $\Delta$ | -.286 | -.140 | .149 | .017 | .327 | .237 | .219 | .076 | .050 | .082 | .078 | .084 | .041 | .018 | .143 | .099 |
| DFCN [48] | DocT-T | .376 | .961 | .123 | .862 | .084 | .679 | .057 | .883 | .220 | .795 | .068 | .791 | .081 | .764 | .102 | .782 |
| | FSTS-T | .195 | .841 | .394 | .917 | .594 | .944 | .334 | .939 | .414 | .910 | .113 | .831 | .182 | .819 | .327 | .889 |
| | **Gain** $\Delta$ | -.181 | -.119 | .271 | .055 | .510 | .265 | .277 | .056 | .194 | .115 | .045 | .040 | .101 | .055 | .225 | .106 |
| PSCC-Net [28] | DocT-T | .325 | .973 | .290 | .846 | .225 | .729 | .091 | .804 | .456 | .848 | .102 | .774 | .261 | .782 | .227 | .787 |
| | FSTS-T | .006 | .535 | .488 | .871 | .651 | .968 | .099 | .766 | .680 | .929 | .307 | .897 | .209 | .716 | .389 | .855 |
| | **Gain** $\Delta$ | -.319 | -.438 | .198 | .025 | .426 | .239 | .008 | -.038 | .224 | .081 | .205 | .123 | -.052 | -.066 | .162 | .068 |
| MVSS-Net [10] | DocT-T | .307 | .721 | .241 | .698 | .196 | .662 | .082 | .728 | .255 | .701 | .104 | .696 | .203 | .698 | .168 | .697 |
| | FSTS-T | .185 | .742 | .491 | .818 | .559 | .878 | .382 | .845 | .445 | .804 | .187 | .755 | .357 | .780 | .386 | .812 |
| | **Gain** $\Delta$ | -.122 | .021 | .250 | .120 | .363 | .215 | .300 | .117 | .189 | .103 | .083 | .059 | .153 | .082 | .218 | .115 |
| TruFor [15] | DocT-T | .516 | .982 | .400 | .901 | .211 | .708 | .185 | .811 | .289 | .811 | .091 | .787 | .214 | .811 | .198 | .785 |
| | FSTS-T | .270 | .868 | .775 | .980 | .683 | .952 | .638 | .984 | .487 | .892 | .190 | .865 | .386 | .866 | .477 | .912 |
| | **Gain** $\Delta$ | -.247 | -.114 | .374 | .079 | .471 | .244 | .453 | .174 | .198 | .082 | .099 | .078 | .172 | .057 | .279 | .127 |
| DTD [32] | DocT-T | .449 | .906 | .129 | .787 | .104 | .658 | .024 | .631 | .164 | .685 | .066 | .670 | .125 | .666 | .097 | .662 |
| | FSTS-T | .121 | .656 | .355 | .822 | .607 | .934 | .115 | .749 | .717 | .934 | .062 | .635 | .225 | .724 | .345 | .795 |
| | **Gain** $\Delta$ | -.328 | -.250 | .226 | .034 | .503 | .276 | .091 | .118 | .553 | .249 | -.004 | -.035 | .100 | .058 | .249 | .133 |
| STFL-Net [45] | DocT-T | .510 | .972 | .370 | .893 | .186 | .679 | .134 | .893 | .306 | .771 | .162 | .794 | .237 | .770 | .205 | .781 |
| | FSTS-T | .138 | .708 | .592 | .921 | .589 | .921 | .451 | .960 | .426 | .872 | .197 | .863 | .332 | .847 | .399 | .892 |
| | **Gain** $\Delta$ | -.372 | -.264 | .222 | .029 | .403 | .242 | .317 | .067 | .100 | .120 | .035 | .069 | .094 | .077 | .194 | .111 |

real-world datasets. This assessment determines whether training exclusively on synthetic data enables the model to perform effectively in practical T-IFL scenarios.

- **Protocol 3: Direct Training and Testing on Real-World Data.** This protocol establishes a baseline by training the model on real-world datasets (e.g., CertificatePS) and evaluating it in both within-dataset and cross-dataset settings. The within-dataset evaluation assesses performance on the same dataset used for training, while the cross-dataset evaluation measures generalization to unseen real-world datasets.

- **Protocol 4: Synthetic Data Pretraining with Real-World Data Fine-Tuning.** The model is first initialized with weights pretrained on synthetic data (from Protocol 1 or 2) and then fine-tuned on real-world data, following the same strategy as Protocol 3. This protocol investigates whether synthetic pretraining improves model performance on real-world T-IFL, particularly when real data are scarce or expensive to obtain.

## 4.2 Comparison with the State-of-the-art Methods

We compare the performance of representative state-of-the-art (SOTA) methods from both N-IFL [7, 48, 28, 10, 15] and T-IFL [45, 32] domains under the four evaluation protocols outlined in Sec. 4.1. All models are trained under identical experimental settings based on the same training and testing splits, following their official implementations and default hyperparameters, with 50 and 25 training epochs for Protocols 1–2 and 3–4, respectively, to ensure fair comparison. Specifically, the compared methods include five N-IFL methods (RRU-Net [7], DFCN [48], PSCC-Net [28], MVSS-Net [10], and TruFor [15]) and two T-IFL methods (DTD [32], STFL-Net [45]). Notably, several of these methods were introduced together with corresponding tampered text image datasets, underscoring their close relevance to our task setting. For example, DFCN introduced a set of synthetic and real-world book-cover tampering datasets, DTD proposed the DocTamper dataset of synthetic document forgeries, and STFL-Net released the STFD dataset of real-world screenshot forgeries.

**Protocol 1.** It can be observed from the left side of Table 2 that models trained on synthetic data (DocT-T, FSTS-T) and tested on the corresponding synthetic data (DocT-S, FSTS-S) demonstrate

Table 3: Pixel-level F1 and AUC performance of T-IFL for Protocols 3 and 4, showing models trained under different strategies and tested on real-world datasets. Each method includes four rows corresponding to different training–testing configurations. The first row (Direct) shows results for models trained and tested directly on real datasets (e.g., CertificatePS) (Protocol 3). The second and third rows (DocT-T and FSTS-T) report results from models pretrained on synthetic datasets and fine-tuned on real datasets (Protocol 4). Subscripts denote performance differences relative to the Direct setting, indicating the impact of synthetic pretraining. The fourth row (**Gain** $\Delta$) highlights performance differences (FSTS-T minus DocT-T). Same highlighting conventions as in Table 2 apply.

| Methods | Train \ Test | FSTS-1.5k F1 | FSTS-1.5k AUC | AFAC F1 | AFAC AUC | CertificatePS F1 | CertificatePS AUC | STFD F1 | STFD AUC | FindIt F1 | FindIt AUC | Average F1 | Average AUC |
|---|---|---|---|---|---|---|---|---|---|---|---|---|---|
| RRU-Net [7] | Direct | .680 | .929 | .075 | .718 | .790 | .971 | .163 | .773 | .250 | .669 | .392 | .812 |
|  | DocT-T | $.459_{-.221}$ | $.857_{-.072}$ | $.084_{.009}$ | $.648_{-.071}$ | $.693_{-.096}$ | $.932_{-.039}$ | $.146_{-.018}$ | $.749_{-.024}$ | $.240_{-.010}$ | $.687_{.019}$ | $.324_{-.067}$ | $.774_{-.037}$ |
|  | FSTS-T | $.687_{.007}$ | $.946_{.018}$ | $.131_{.056}$ | $.815_{.097}$ | $.819_{.029}$ | $.973_{.002}$ | $.177_{.014}$ | $.798_{.025}$ | $.261_{.011}$ | $.714_{.045}$ | $.415_{.023}$ | $.849_{.037}$ |
|  | **Gain** $\Delta$ | .229 | .090 | .047 | .167 | .126 | .041 | .032 | .049 | .021 | .027 | .091 | .075 |
| DFCN [48] | Direct | .547 | .901 | .065 | .693 | .699 | .953 | .156 | .756 | .153 | .663 | .324 | .793 |
|  | DocT-T | $.453_{-.094}$ | $.849_{-.052}$ | $.076_{.011}$ | $.701_{.008}$ | $.652_{-.047}$ | $.927_{-.026}$ | $.134_{-.021}$ | $.727_{-.029}$ | $.220_{.067}$ | $.732_{.069}$ | $.307_{-.017}$ | $.787_{-.006}$ |
|  | FSTS-T | $.730_{.183}$ | $.958_{.057}$ | $.064_{-.002}$ | $.758_{.066}$ | $.844_{.145}$ | $.987_{.034}$ | $.163_{.008}$ | $.767_{.011}$ | $.201_{.048}$ | $.703_{.040}$ | $.400_{.076}$ | $.835_{.042}$ |
|  | **Gain** $\Delta$ | .277 | .110 | -.012 | .057 | .192 | .060 | .029 | .040 | -.019 | -.028 | .093 | .048 |
| PSCC-Net [28] | Direct | .684 | .940 | .064 | .733 | .862 | .992 | .157 | .758 | .254 | .659 | .404 | .816 |
|  | DocT-T | $.690_{.006}$ | $.944_{.004}$ | $.074_{.010}$ | $.712_{-.020}$ | $.855_{-.007}$ | $.992_{.000}$ | $.184_{.027}$ | $.750_{-.008}$ | $.277_{.023}$ | $.721_{.062}$ | $.416_{.012}$ | $.824_{.008}$ |
|  | FSTS-T | $.707_{.023}$ | $.938_{-.002}$ | $.075_{.011}$ | $.698_{-.034}$ | $.865_{.003}$ | $.993_{.001}$ | $.191_{.034}$ | $.784_{.026}$ | $.251_{-.003}$ | $.740_{.081}$ | $.418_{.013}$ | $.831_{.014}$ |
|  | **Gain** $\Delta$ | .017 | -.006 | .001 | -.014 | .010 | .001 | .007 | .034 | -.026 | .019 | .002 | .007 |
| MVSS-Net [10] | Direct | .674 | .914 | .053 | .598 | .871 | .958 | .128 | .661 | .298 | .659 | .405 | .758 |
|  | DocT-T | $.651_{-.023}$ | $.910_{-.004}$ | $.043_{-.010}$ | $.516_{-.082}$ | $.858_{-.014}$ | $.971_{.013}$ | $.139_{.011}$ | $.656_{-.005}$ | $.339_{.041}$ | $.734_{.076}$ | $.406_{.001}$ | $.758_{.000}$ |
|  | FSTS-T | $.710_{.036}$ | $.939_{.025}$ | $.082_{.029}$ | $.662_{.064}$ | $.876_{.005}$ | $.973_{.016}$ | $.143_{.015}$ | $.715_{.055}$ | $.362_{.064}$ | $.745_{.086}$ | $.434_{.030}$ | $.807_{.049}$ |
|  | **Gain** $\Delta$ | .059 | .029 | .039 | .146 | .018 | .002 | .004 | .060 | .022 | .011 | .029 | .049 |
| TruFor [15] | Direct | .758 | .961 | .137 | .825 | .844 | .985 | .172 | .803 | .386 | .850 | .459 | .885 |
|  | DocT-T | $.736_{-.023}$ | $.952_{-.009}$ | $.166_{.029}$ | $.817_{-.008}$ | $.839_{-.005}$ | $.982_{-.003}$ | $.166_{-.006}$ | $.786_{-.017}$ | $.358_{-.028}$ | $.838_{-.012}$ | $.453_{-.007}$ | $.875_{-.010}$ |
|  | FSTS-T | $.784_{.026}$ | $.972_{.011}$ | $.238_{.100}$ | $.869_{.044}$ | $.857_{.013}$ | $.988_{.003}$ | $.192_{.020}$ | $.828_{.025}$ | $.418_{.032}$ | $.847_{-.003}$ | $.498_{.038}$ | $.901_{.016}$ |
|  | **Gain** $\Delta$ | .049 | .020 | .072 | .052 | .018 | .006 | .026 | .041 | .060 | .009 | .045 | .026 |
| DTD [32] | Direct | .607 | .922 | .046 | .627 | .876 | .982 | .087 | .733 | .206 | .695 | .364 | .792 |
|  | DocT-T | $.617_{.010}$ | $.926_{.004}$ | $.037_{-.010}$ | $.630_{.003}$ | $.887_{.011}$ | $.982_{.000}$ | $.095_{.008}$ | $.705_{-.028}$ | $.257_{.051}$ | $.724_{.029}$ | $.378_{.014}$ | $.793_{.002}$ |
|  | FSTS-T | $.633_{.027}$ | $.932_{.010}$ | $.051_{.005}$ | $.650_{.023}$ | $.893_{.017}$ | $.982_{.000}$ | $.112_{.025}$ | $.743_{.010}$ | $.294_{.088}$ | $.743_{.048}$ | $.397_{.032}$ | $.810_{.018}$ |
|  | **Gain** $\Delta$ | .017 | .006 | .015 | .020 | .005 | .000 | .017 | .038 | .037 | .019 | .018 | .017 |
| STFL-Net [45] | Direct | .658 | .935 | .094 | .770 | .858 | .988 | .141 | .765 | .318 | .774 | .414 | .846 |
|  | DocT-T | $.665_{.007}$ | $.942_{.008}$ | $.110_{.016}$ | $.837_{.068}$ | $.848_{-.011}$ | $.985_{-.003}$ | $.146_{.005}$ | $.778_{.013}$ | $.329_{.011}$ | $.830_{.056}$ | $.420_{.006}$ | $.874_{.028}$ |
|  | FSTS-T | $.727_{.069}$ | $.961_{.026}$ | $.135_{.041}$ | $.856_{.086}$ | $.877_{.018}$ | $.989_{.001}$ | $.165_{.024}$ | $.799_{.034}$ | $.337_{.019}$ | $.839_{.066}$ | $.448_{.034}$ | $.889_{.043}$ |
|  | **Gain** $\Delta$ | .062 | .019 | .025 | .019 | .029 | .004 | .019 | .021 | .008 | .009 | .029 | .014 |

better performance, validating the effectiveness of synthetic data training. However, similar to the approach in DocTamper [32], although this setup yields favorable results, it has limited practical value due to the high similarity in tampering distributions between training and testing data, which hinders a comprehensive evaluation of both the data quality and the model's generalization ability.

**Protocol 2.** It can be observed from the right side of Table 2 that models trained on FSTS-T consistently outperform those trained on DocT-T when evaluated on real-world datasets, indicating that FSTS-T provides more effective training data for real-world generalization. The average F1 gain across all methods exceeds 14%, with some models (e.g., DFCN, MVSS-Net, DTD, TruFor) achieving gains of over 21%. However, some methods still exhibit suboptimal performance on specific datasets. For example, PSCC-Net performs poorly on AFAC, and DTD underperforms on STFD, possibly due to their limited ability to extract discriminative features from low-texture text images. These results demonstrate the effectiveness of our FSTS strategy in generating synthetic data that enhances cross-domain generalization across diverse real-world T-IFL scenarios.

**Protocol 3.** As shown in the first row (Direct) of each method in Table 3, models trained on the real-world dataset (CertificatePS) achieve solid performance in within-dataset testing. However, their performance drops significantly in cross-dataset scenarios (e.g., AFAC, STFD, FindIt), indicating the limited generalization ability of models trained solely on real-world data. Furthermore, compared with the results in Table 2, almost all models trained on real-world data consistently underperform our proposed FSTS-T counterparts in cross-dataset evaluations on AFAC and STFD. We also observe that models trained on DocT-T achieve similarly limited results on these datasets, comparable to those trained on real-world annotations. These findings highlight the crucial role of our proposed high-quality synthetic datasets, i.e., FSTS-T, which provide more diverse and generalizable supervisory signals than limited real-world annotations.

**Protocol 4.** As illustrated in the second and third rows in Table 3, almost all methods benefit from pretraining on FSTS-T followed by fine-tuning on CertificatePS, yielding consistent performance gains despite the limitations noted in Protocol 2 (e.g., PSCC-Net's poor performance on AFAC). In contrast, when methods are pretrained on DocT-T and then fine-tuned on CertificatePS, many exhibit negative gains on multiple real datasets, indicating that FSTS-T outperforms DocT-T in improving model generalization, particularly when real data is limited. These results further confirm the superiority of FSTS-T over conventional synthetic datasets as a pretraining source for enhancing real-world T-IFL performance.

## 5 Conclusion

In this paper, we present Fourier Series-based Tampering Synthesis (FSTS), a structured and interpretable framework for generating realistic tampered text images by modeling the invisible distribution of real-world tampering parameters. To achieve this, we first design a structured pipeline that collects 16,750 real-world tampering instances across five representative tampering types, capturing fine-grained editing traces from 67 human participants. We then analyze the collected data and identify recurring behavioral patterns at both the individual and population levels, which serve as the foundation for our hierarchical distribution modeling framework inspired by the Fourier series. By sampling operation–parameter configurations and their learned frequencies from this model, FSTS synthesizes diverse and realistic tampered samples that better reflect the complexity of real-world forgeries. Extensive experiments under four evaluation protocols confirm the superiority of FSTS-synthesized data in enhancing model generalization across various real-world T-IFL benchmarks.

## 6 Acknowledgments

This work was primarily supported by a self-funded project led by Zeqin Yu, and partially supported by the following funding sources: the National Natural Science Foundation of China under Grant U23B2022 and Grant U22A2030, the Guangdong Major Project of Basic and Applied Basic Research under Grant 2023B0303000010, the National Natural Science Foundation of China under Grant 62202507, the Natural Science Foundation of Guangdong Province under Grant 2025A1515012830, and the Guangzhou Municipal Government-University (Institute) Enterprises Jointly Founded Project under Grant 2025A03J3123, and the China Scholarship Council under File No. 202506380134.

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

# A    Overview of Existing Tampered Text Image Datasets

## A.1    Summary of Existing Datasets

We provide a comparison of several representative tampered text image datasets for T-IFL, as shown in Table 4. These datasets differ in terms of their tampering sources, sample count, supported tampering types, public accessibility, and release year. Real-world datasets, while valuable, are limited in number and often private or not publicly accessible. As a result, obtaining diverse, real-world tampered text images for model training and evaluation remains a significant challenge. In contrast, synthetic datasets offer a larger sample size but are limited in their ability to reflect the invisible distribution of tampering parameters observed in real-world data (which we analyze in the next subsection). The following itemized list provides an overview of the datasets used in this paper. Unless otherwise specified, all datasets follow a consistent preprocessing procedure [45, 32]: images are cropped into $512 \times 512$ patches to standardize the dataset for evaluation purposes.

- **DocTamper** is a synthetic dataset applied to various scanned documents, such as contracts, receipts, invoices, and books. It follows rule-based pipelines that apply predefined tampering types to text images, aiming to generate large-scale datasets. All images in the dataset are $512 \times 512$ image patches. The training set consists of 120,000 tampered images, and the test set is divided into three subsets: DocTamper-Test (30,000), DocTamper-FCD (2,000), and DocTamper-SCD (18,000). Notably, both the training and test datasets are synthesized using the same generation process, resulting in similar tampering distributions.

- **CertificatePS** is a certificate dataset consisting of 4,840 tampered images captured under both indoor and outdoor settings using multiple devices. The original image resolutions range from $640 \times 852$ to $6,944 \times 9,248$ pixels. We randomly select 1,000 images and crop them into $512 \times 512$ patches, resulting in 9,210 patches used for evaluation purposes.

- **AFAC** is a competition dataset constructed from diverse sources such as photographed documents, receipts, scanned documents, and street view images. It contains 5,632 tampered images, created using techniques like copy-move, splicing, and removal, though the tampering types are not explicitly labeled. After cropping, the dataset includes 15,387 patches, which are used for evaluation purposes.

- **STFD** is the first dataset dedicated to smartphone screenshot images, encompassing common scenarios in daily life such as chat records, money transfer receipts, and news pages. Compared to other types of text images, screenshot images are created by directly capturing screen pixels, resulting in simpler background textures but posing greater challenges for IFL tasks. The dataset includes 4,094 images and 30,269 patches after cropping.

- **FindIt** is a fraud detection dataset featuring receipts from franchises, brand stores, and independent shops. It includes common fraud types such as price alterations and product modifications, along with challenges like folds, stains, and faded ink, making it a valuable benchmark for forgery detection research. The dataset includes 240 tampered images and 968 tampered patches after cropping.

- **TIC13** is a synthetic scene text image dataset containing images tampered by the GAN-based editing model [43]. The dataset includes 986 images and 1,768 patches after cropping.

- **T-SROIE** is a synthetic dataset similar to TIC13, using the same tampering methods but applied to small ticket-like images such as receipts. The dataset includes 462 images and 4,343 patches after cropping.

- **OSTF** is a synthetic scene text image dataset, which contains natural scene texts tampered with using eight different GAN and Diffusion models-based text editing techniques. The dataset includes 1,980 images and 6,354 patches after cropping.

## A.2    Limitations of Existing Synthetic Datasets

Existing synthetic datasets often employ highly repetitive tampering pipelines, which result in narrow and less diverse invisible distributions of tampering parameters. As shown in Fig. 5(1-4), even

Table 4: Comparison of representative tampered text image datasets for T-IFL. For each dataset, we report its tampering source (synthetic or real-world), the number of tampered samples available in image-level and patch-level formats, the supported tampering types, public accessibility, and the year of release. Tampering type abbreviations are as follows: Com (Copy-move), Spl (Splicing), Rem (Removal), Ins (Insertion), Rep (Replacement).

| Dataset | Tampering Source | | Sample Count | | Tampering Type | Publicly | Year |
|---|---|---|---|---|---|---|---|
| | Synthetic | Real-world | Image | Patch | | | |
| FindIt [2] | - | ✓ | 240 | 968 | Com, Spl, Rep | ✓ | 2017 |
| PS-arbitrary [48] | - | ✓ | 1,000 | - | Spl | ✗ | 2021 |
| PS-boundary [48] | - | ✓ | 1,000 | - | Spl | ✗ | 2021 |
| PS-scripted [48] | ✓ | - | 14,581 | - | Spl | ✗ | 2021 |
| TIC13 [41] | ✓ | - | 986 | 1,768 | Rep | ✓ | 2022 |
| T-SROIE [42] | ✓ | - | 462 | 4,343 | Rep | ✓ | 2022 |
| STFD [45] | - | ✓ | 4,094 | 30,269 | Com, Spl, Rem, Ins, Rep | ✓ | 2023 |
| CertificatePS [49] | - | ✓ | 4,840 | 9,210 | Com, Spl, Rem, Ins, Rep | ✓ | 2023 |
| DocTamper [32] | ✓ | - | 170,000 | 170,000 | Com, Spl, Rep | ✓ | 2023 |
| AFAC [1] | - | ✓ | 5,632 | 15,387 | - | ✓ | 2023 |
| OSTF [33] | ✓ | - | 1,980 | 6,354 | Rep | ✓ | 2025 |
| Ours | ✓ | ✓ | 294,182 | 294,182 | Com, Spl, Rem, Ins, Rep | ✓ | 2025 |

visually different samples tend to follow similar operation-parameter patterns. In contrast, as shown in Fig. 5(5-8), our collected real-world replacement examples exhibit rich combinations of operations such as insertion, stroke, blur, and color manipulation, motivating the need to explicitly model tampering parameter distributions when synthesizing training data. This motivates our approach to explicitly model the diversity of tampering parameters, ensuring a more representative synthesis of tampered data.

## B Details of Our Dataset Construction

### B.1 Tampering Parameter Collection and Modeling

In this section, we present the tampering parameters modeled in Sec. 3.3 of the main paper for five representative tampering types $\phi_k$: Copy-move, Splicing, Removal, Insertion, and Replacement. These parameters are summarized in Tables 8– 12. For each tampering type $\phi_k$, we categorize the operations into two main parts: main processing and post-processing[2]. These are further organized into representative steps, such as region sampling, geometric transformation, and visual trace concealment for the Copy-move tampering type, as detailed in Table 8. In total, $I = 67$ individuals participated in the tampering collection process, each performing approximately $J \approx 50$ operations per tampering type (about 250 in total), resulting in 16,750 real-world instances across five representative tampering types. In each table, the **Parameter Type** and **Parameter Value** columns correspond to the tampering parameters $t_k$ and $a_k$, while the **Frequency** column represents the $a_k$ values, indicating the frequency of each operation variant's use during the synthesis process, as defined and modeled in Sec. 3.3.

### B.2 Synthetic Image Generation

In this section, we describe the synthetic image generation process, which is illustrated in Fig. 4. The process begins with a target image $I^0$ (Fig. 4(I)(a)) and applies predefined tampering types $\phi_k$, utilizing the tampering parameters $t_k$ and frequency weights $a_k$ (Fig. 4(I)(f)). The key steps in the tampering process are as follows:

- **Text Region Manipulation**: The coordinates of the text region are first obtained using an existing OCR tool (e.g., PaddleOCR [4]), and then the text region of the target image is manipulated according to the tampering type selected, using Photoshop. This manipulation

---

[2]For the post-processing operations, we scaled the frequency values by a factor of 0.3 to prevent overly complex tampered samples that could hinder model training. This scaling was necessary because, while all parameters in post-processing have a chance of being used, only a subset is selected in practice. Using the original frequencies could lead to impractically complex tampering samples. This approach is consistent across all five tampering types.

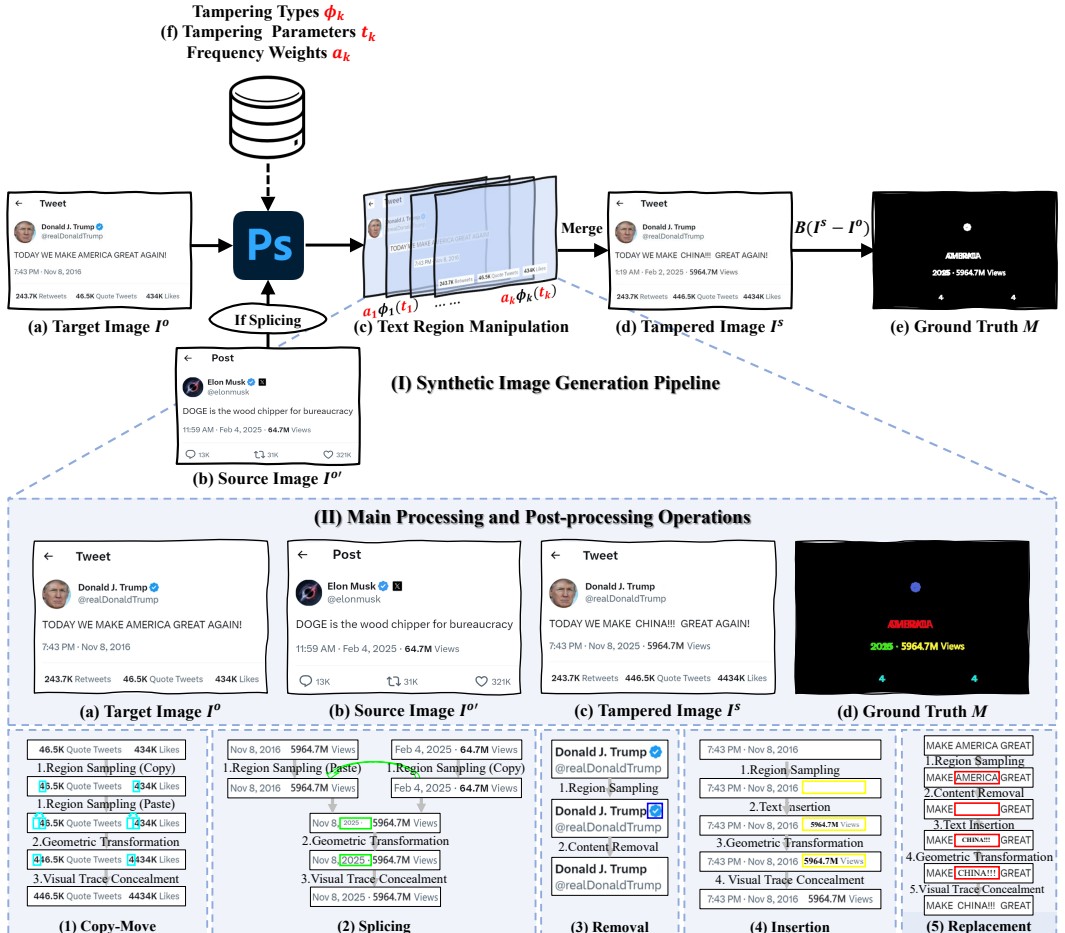

Figure 4: Synthetic Tampered Text Image Generation Pipeline with Parameters Modeled by FSTS. (I) The overall pipeline takes a target image and synthesizes a tampered version along with its corresponding ground-truth mask, using tampering types, parameter configurations, and frequency weights modeled by our proposed FSTS framework. (II) This panel zooms into the tampering step in (I), detailing the main and post-processing operations for five representative tampering types. We use consistent color coding to distinguish different tampering types in both the ground-truth mask (II)(d) and the operation detail panels (II)(1-5): (1) Copy-move (copy and move text within the same image), (2) Splicing (pastes text from a source image to a target image), (3) Removal (erases text followed by in-painting), (4) Insertion (inserts forged text into blank regions), (5) Replacement (generates forged text to replace original text).

process is guided by the parameters $t_k$ and frequency weights $a_k$, with main processing and post-processing operations, as defined in the five tables (Tables 8– 12). The manipulation is carried out on the text region (Fig. 4(I)(c)) and is automated using a JavaScript-based script, which handles the complex operations on the text regions.

- **Layer Merging**: Once the text region manipulation is completed, multiple tampered image layers are merged to form the final tampered image $I^s$, as shown in Fig. 4(I)(d).

- **Mask Generation**: To create a ground truth mask, the difference between the tampered image $I^s$ and the original image $I^0$ is calculated. This difference is then thresholded using a binarization function $B(I^s - I^0)$, resulting in the mask $M$ (Fig. 4(I)(e)), which highlights the tampered regions of the image. The tampered regions are marked as 1, while the non-tampered regions are marked as 0.

As illustrated in Fig. 4(II), we further provide detailed descriptions of five representative tampering types, each consisting of both main and post-processing steps. Sub-panels (1)–(5) show concrete examples, and the tampered regions are highlighted with distinct colors in the ground-truth mask $M$.

- **Copy-move:** As shown in Fig. 4(II)(1), a digit "4" is selected from the target image $I^o$ (*1. Region Sampling (Copy)*) and pasted into a nearby location (*1. Region Sampling (Paste)*). The pasted region is then adjusted (*2. Geometric Transformation*) and refined (*3. Visual Trace Concealment*). The resulting tampered image $I^s$ contains an additional "4", highlighted in light blue in the ground-truth mask $M$.

- **Splicing:** As shown in Fig. 4(II)(2), the digit "2025" is sampled from a source image $I^{o'}$ (*1. Region Sampling (Copy)*) and pasted over "2016" in the target image $I^o$ (*1. Region Sampling (Paste)*). The pasted region is then adjusted (*2. Geometric Transformation*) and refined (*3. Visual Trace Concealment*). The resulting tampered image $I^s$ shows "2016" replaced with "2025". The tampered area is highlighted in green in the ground-truth mask $M$.

- **Removal:** As shown in Fig. 4(II)(3), a certification mark is selected from the target image $I^o$ (*1. Region Sampling*) and erased (*2. Content Removal*). The resulting tampered image $I^s$ no longer contains the mark. The tampered area is highlighted in dark blue in the ground-truth mask $M$.

- **Insertion:** As shown in Fig. 4(II)(4), a blank region in the image $I^o$ is selected (*1. Region Sampling*), and new text such as "5964.7M Views" is inserted (*2. Text Insertion*). The text is then adjusted (*3. Geometric Transformation*) and refined (*4. Visual Trace Concealment*). The resulting tampered image $I^s$ contains the newly added text. The tampered region is highlighted in yellow in the ground-truth mask $M$.

- **Replacement:** As shown in Fig. 4(II)(5), the text "AMERICA" is selected from the target image $I^o$ (*1. Region Sampling*) and erased (*2. Content Removal*). New text, "CHINA!!!", is inserted in the same location (*3. Text Insertion*), then adjusted (*4. Geometric Transformation*) and refined (*5. Visual Trace Concealment*). The resulting tampered image $I^s$ shows "AMERICA" replaced with "CHINA!!!". The tampered region is highlighted in red in the ground-truth mask $M$.

## B.3 Dataset Variants

In this paper, as shown in the last row of Table 4, we present the FSTS dataset, which ultimately contains 294,182 images, including both real-world and synthetic datasets. **To the best of our knowledge, this is currently the largest tampered text image dataset.** The dataset consists of the following components:

- **FSTS-T**: The FSTS-T dataset consists of 50,000 synthetic images used for training. These images are generated using the FSTS strategy, with the original text images [8, 17, 26, 37] following the DocTamper-Train (DocT-T) protocol [32], ensuring that the visible distributions in the dataset are similar, thus emphasizing the comparison of the invisible distribution differences between the two datasets.

- **FSTS-S**: The FSTS-S dataset consists of 5,705 tampered images, generated using the proposed FSTS strategy on the Huawei Cloud document dataset [18]. This dataset is consistent with the second cross-domain setting in DocTamper (DocT-S), serving as a validation set for synthetic data, although we believe the practical significance of validation under such protocols is limited.

- **FSTS-1.5k**: The FSTS-1.5k dataset consists of approximately 1.5k (specifically 1,488) tampered text images excluded from the parameter modeling in FSTS. These images were tampered by five independent tamperers who were not involved in the FSTS framework, ensuring diversity for evaluating generalization performance.

- **FSTS-v2**: Additionally, we provide the FSTS-v2 dataset, which consists of 236,989 synthetic samples designed to provide a large number of samples for optional pretraining. These samples are generated in the same way as FSTS-T, but using different original text images sourced from [44, 31, 25, 16, 35, 36, 29, 19, 9].

Table 5: Pixel-level F1 and AUC performance of T-IFL under Protocol 2, extended to include three additional synthetic datasets: TIC13 [41], T-SROIE [42], and OSTF [33]. Each method includes five rows corresponding to different training–testing configurations. The first row reports results for models trained on FSTS-T (**Ours**). The next four rows report results for models trained on four existing synthetic datasets used for comparison: DocT-T, T-SROIE, TIC13, and OSTF. Performance gains (**others minus FSTS-T**) are shown as subscripts in each cell. Positive gains are highlighted in red, and negative gains in blue.

| Methods | Test / Train | FSTS-1.5k F1 | AUC | AFAC F1 | AUC | Certificate F1 | AUC | STFD F1 | AUC | FindIt F1 | AUC | Average F1 | AUC |
|---|---|---|---|---|---|---|---|---|---|---|---|---|---|
| RRU-Net [7] | **FSTS-T** | .541 | .933 | .307 | .874 | .433 | .863 | .177 | .855 | .252 | .793 | .342 | .864 |
| | DocT-T | $.215_{-.327}$ | $.696_{-.237}$ | $.088_{-.219}$ | $.798_{-.076}$ | $.383_{-.050}$ | $.782_{-.082}$ | $.099_{-.078}$ | $.772_{-.084}$ | $.211_{-.041}$ | $.776_{-.018}$ | $.199_{-.143}$ | $.765_{-.099}$ |
| | T-SROIE | $.125_{-.417}$ | $.610_{-.323}$ | $.140_{-.167}$ | $.654_{-.220}$ | $.015_{-.419}$ | $.534_{-.329}$ | $.037_{-.139}$ | $.502_{-.353}$ | $.011_{-.241}$ | $.500_{-.293}$ | $.066_{-.276}$ | $.560_{-.304}$ |
| | TIC13 | $.182_{-.359}$ | $.660_{-.273}$ | $.129_{-.178}$ | $.795_{-.079}$ | $.315_{-.118}$ | $.723_{-.140}$ | $.143_{-.033}$ | $.759_{-.097}$ | $.167_{-.085}$ | $.693_{-.100}$ | $.187_{-.155}$ | $.726_{-.138}$ |
| | OSTF | $.255_{-.287}$ | $.682_{-.251}$ | $.231_{-.076}$ | $.862_{-.011}$ | $.327_{-.107}$ | $.750_{-.113}$ | $.119_{-.058}$ | $.736_{-.119}$ | $.142_{-.110}$ | $.698_{-.096}$ | $.215_{-.127}$ | $.746_{-.118}$ |
| DFCN [48] | **FSTS-T** | .594 | .944 | .334 | .939 | .414 | .910 | .113 | .831 | .182 | .819 | .327 | .889 |
| | DocT-T | $.084_{-.510}$ | $.679_{-.265}$ | $.057_{-.277}$ | $.883_{-.056}$ | $.220_{-.194}$ | $.795_{-.115}$ | $.068_{-.045}$ | $.791_{-.040}$ | $.081_{-.101}$ | $.764_{-.055}$ | $.102_{-.225}$ | $.782_{-.106}$ |
| | T-SROIE | $.065_{-.529}$ | $.619_{-.325}$ | $.107_{-.227}$ | $.781_{-.158}$ | $.016_{-.398}$ | $.606_{-.305}$ | $.023_{-.090}$ | $.503_{-.328}$ | $.014_{-.168}$ | $.566_{-.253}$ | $.045_{-.282}$ | $.615_{-.274}$ |
| | TIC13 | $.172_{-.422}$ | $.631_{-.313}$ | $.134_{-.200}$ | $.810_{-.129}$ | $.281_{-.133}$ | $.735_{-.175}$ | $.138_{.026}$ | $.764_{-.068}$ | $.171_{-.012}$ | $.725_{-.094}$ | $.179_{-.148}$ | $.733_{-.156}$ |
| | OSTF | $.164_{-.430}$ | $.674_{-.270}$ | $.094_{-.241}$ | $.832_{-.107}$ | $.276_{-.138}$ | $.776_{-.134}$ | $.127_{.014}$ | $.763_{-.068}$ | $.146_{-.037}$ | $.723_{-.096}$ | $.161_{-.166}$ | $.754_{-.135}$ |
| PSCC-Net [28] | **FSTS-T** | .651 | .968 | .099 | .766 | .680 | .929 | .307 | .897 | .209 | .716 | .389 | .855 |
| | DocT-T | $.225_{-.426}$ | $.729_{-.239}$ | $.091_{-.008}$ | $.804_{.038}$ | $.456_{-.224}$ | $.848_{-.081}$ | $.102_{-.205}$ | $.774_{-.123}$ | $.261_{.052}$ | $.782_{.066}$ | $.227_{-.162}$ | $.787_{-.068}$ |
| | T-SROIE | $.184_{-.467}$ | $.664_{-.304}$ | $.205_{.106}$ | $.697_{-.070}$ | $.135_{-.545}$ | $.658_{-.271}$ | $.026_{-.281}$ | $.543_{-.354}$ | $.063_{-.146}$ | $.621_{-.095}$ | $.123_{-.266}$ | $.637_{-.219}$ |
| | TIC13 | $.186_{-.465}$ | $.695_{-.273}$ | $.128_{.029}$ | $.786_{.020}$ | $.335_{-.346}$ | $.757_{-.172}$ | $.132_{-.175}$ | $.741_{-.156}$ | $.189_{-.020}$ | $.722_{.006}$ | $.194_{-.195}$ | $.740_{-.115}$ |
| | OSTF | $.225_{-.426}$ | $.698_{-.270}$ | $.166_{.067}$ | $.857_{.091}$ | $.380_{-.300}$ | $.792_{-.137}$ | $.134_{-.173}$ | $.754_{-.143}$ | $.203_{-.006}$ | $.713_{-.003}$ | $.222_{-.168}$ | $.763_{-.092}$ |
| MVSS-Net [10] | **FSTS-T** | .559 | .878 | .382 | .845 | .445 | .804 | .187 | .755 | .357 | .780 | .386 | .812 |
| | DocT-T | $.196_{-.363}$ | $.662_{-.215}$ | $.082_{-.300}$ | $.728_{-.117}$ | $.255_{-.189}$ | $.701_{-.103}$ | $.104_{-.083}$ | $.696_{-.059}$ | $.203_{-.153}$ | $.698_{-.082}$ | $.168_{-.218}$ | $.697_{-.115}$ |
| | T-SROIE | $.079_{-.480}$ | $.512_{-.366}$ | $.101_{-.281}$ | $.644_{-.201}$ | $.042_{-.403}$ | $.489_{-.315}$ | $.052_{-.135}$ | $.467_{-.288}$ | $.031_{-.325}$ | $.537_{-.243}$ | $.061_{-.325}$ | $.530_{-.283}$ |
| | TIC13 | $.179_{-.380}$ | $.647_{-.230}$ | $.128_{-.254}$ | $.803_{-.042}$ | $.323_{-.121}$ | $.707_{-.098}$ | $.138_{-.049}$ | $.726_{-.029}$ | $.190_{-.167}$ | $.710_{-.071}$ | $.192_{-.194}$ | $.718_{-.094}$ |
| | OSTF | $.232_{-.327}$ | $.689_{-.188}$ | $.243_{-.139}$ | $.835_{-.010}$ | $.263_{-.181}$ | $.708_{-.096}$ | $.093_{-.093}$ | $.654_{-.101}$ | $.143_{-.214}$ | $.656_{-.124}$ | $.195_{-.191}$ | $.708_{-.104}$ |
| TruFor [15] | **FSTS-T** | .683 | .952 | .638 | .984 | .487 | .892 | .190 | .865 | .386 | .868 | .477 | .912 |
| | DocT-T | $.211_{-.471}$ | $.708_{-.244}$ | $.185_{-.453}$ | $.811_{-.174}$ | $.289_{-.198}$ | $.811_{-.082}$ | $.091_{-.099}$ | $.787_{-.078}$ | $.214_{-.172}$ | $.811_{-.057}$ | $.198_{-.279}$ | $.785_{-.127}$ |
| | T-SROIE | $.086_{-.597}$ | $.591_{-.361}$ | $.227_{-.411}$ | $.755_{-.230}$ | $.037_{-.450}$ | $.566_{-.327}$ | $.031_{-.159}$ | $.519_{-.346}$ | $.024_{-.361}$ | $.590_{-.279}$ | $.081_{-.396}$ | $.604_{-.308}$ |
| | TIC13 | $.192_{-.491}$ | $.714_{-.238}$ | $.142_{-.497}$ | $.846_{-.139}$ | $.326_{-.161}$ | $.763_{-.129}$ | $.156_{-.034}$ | $.784_{-.081}$ | $.197_{-.189}$ | $.768_{-.100}$ | $.202_{-.274}$ | $.775_{-.137}$ |
| | OSTF | $.234_{-.448}$ | $.731_{-.220}$ | $.424_{-.215}$ | $.913_{-.072}$ | $.310_{-.177}$ | $.769_{-.123}$ | $.114_{-.076}$ | $.735_{-.130}$ | $.213_{-.173}$ | $.788_{-.081}$ | $.259_{-.218}$ | $.787_{-.125}$ |
| DTD [32] | **FSTS-T** | .607 | .934 | .115 | .749 | .717 | .934 | .062 | .635 | .225 | .724 | .345 | .795 |
| | DocT-T | $.104_{-.503}$ | $.658_{-.276}$ | $.024_{-.091}$ | $.631_{-.118}$ | $.164_{-.553}$ | $.685_{-.249}$ | $.066_{.004}$ | $.670_{.035}$ | $.125_{-.100}$ | $.666_{-.058}$ | $.097_{-.249}$ | $.662_{-.133}$ |
| | T-SROIE | $.081_{-.527}$ | $.661_{-.273}$ | $.027_{-.088}$ | $.653_{-.096}$ | $.003_{-.714}$ | $.558_{-.376}$ | $.006_{-.056}$ | $.562_{-.073}$ | $.010_{-.215}$ | $.518_{-.206}$ | $.025_{-.320}$ | $.591_{-.205}$ |
| | TIC13 | $.151_{-.456}$ | $.666_{-.268}$ | $.120_{.005}$ | $.794_{.045}$ | $.251_{-.466}$ | $.715_{-.219}$ | $.144_{.082}$ | $.754_{.119}$ | $.171_{-.054}$ | $.723_{-.001}$ | $.168_{-.178}$ | $.731_{-.065}$ |
| | OSTF | $.183_{-.424}$ | $.699_{-.235}$ | $.098_{-.017}$ | $.726_{-.023}$ | $.245_{-.472}$ | $.760_{-.174}$ | $.067_{.005}$ | $.684_{.049}$ | $.126_{-.099}$ | $.688_{-.036}$ | $.144_{-.201}$ | $.711_{-.084}$ |
| STFL-Net [45] | **FSTS-T** | .589 | .921 | .451 | .960 | .426 | .872 | .197 | .863 | .332 | .847 | .399 | .892 |
| | DocT-T | $.186_{-.403}$ | $.679_{-.242}$ | $.134_{-.317}$ | $.893_{-.067}$ | $.306_{-.120}$ | $.771_{-.100}$ | $.162_{-.035}$ | $.794_{-.069}$ | $.237_{-.094}$ | $.770_{-.077}$ | $.205_{-.194}$ | $.781_{-.111}$ |
| | T-SROIE | $.045_{-.545}$ | $.715_{-.206}$ | $.137_{-.314}$ | $.906_{-.054}$ | $.014_{-.413}$ | $.735_{-.137}$ | $.106_{-.091}$ | $.483_{-.380}$ | $.015_{-.316}$ | $.746_{-.101}$ | $.063_{-.336}$ | $.717_{-.176}$ |
| | TIC13 | $.184_{-.406}$ | $.653_{-.268}$ | $.142_{-.310}$ | $.870_{-.090}$ | $.281_{-.145}$ | $.695_{-.177}$ | $.068_{-.129}$ | $.338_{-.525}$ | $.199_{-.133}$ | $.741_{-.105}$ | $.174_{-.224}$ | $.659_{-.233}$ |
| | OSTF | $.228_{-.362}$ | $.680_{-.240}$ | $.207_{-.244}$ | $.790_{-.170}$ | $.253_{-.174}$ | $.684_{-.188}$ | $.068_{-.129}$ | $.338_{-.525}$ | $.202_{-.130}$ | $.737_{-.110}$ | $.191_{-.208}$ | $.646_{-.247}$ |

# C  Additional Experiments

## C.1  Extended Protocol 2 Evaluation

To further validate the generalization ability of our proposed FSTS-T across a broader set of synthetic-to-real domain shifts, we extend Protocol 2 to include three additional synthetic datasets: TIC13 [41], T-SROIE [42], and OSTF [33]. As shown in Table 5, for each baseline model, we compare the performance of FSTS-T against four existing synthetic training datasets on five real-world test sets. Each cell reports the pixel-level F1 and AUC scores, with the relative difference (**others minus FSTS-T**) annotated as a subscript. Red subscripts indicate that the compared method outperforms FSTS-T (i.e., FSTS-T performs worse), while blue subscripts indicate that it underperforms FSTS-T (i.e., FSTS-T performs better).

Across all tested models, FSTS-T consistently achieves the highest average performance on both F1 and AUC metrics. For T-IFL methods, STFL-Net trained on FSTS-T outperforms its counterparts trained on DocT-T, TIC13, T-SROIE, and OSTF, with average gains of over 24.1% in F1 and 19.2% in AUC. For N-IFL methods, TruFor also shows substantial improvement when trained on FSTS-T, achieving an average F1 gain of 29.2% and AUC gain of 17.4% compared with models trained on other synthetic datasets. Furthermore, for DTD, MVSS-Net, PSCC-Net, DFCN, and RRU-Net, FSTS-T provides consistent and significant F1 improvements of over 23.7%, 23.2%, 19.8%, 20.5%, and 17.5%, respectively, further confirming the strong generalization and versatility of our synthetic

Table 6: Pixel-level F1 and AUC performance of image forgery localization for Protocols 3 and 4, showing models trained under different strategies and tested on real-world datasets. Each method has 4 rows corresponding to the training and testing configurations below. The first row (Direct) shows results for models trained and tested directly on real datasets (e.g., STFD ), corresponding to Protocol 3. The second and third rows (DocT-T and FSTS-T) show results for models pretrained on synthetic datasets, then fine-tuned and tested on real datasets, corresponding to Protocol 4. The subscripts in these rows indicate differences from the first row (Direct), reflecting the impact of synthetic pre-training. The fourth row (**Gain** Δ) highlights performance differences (FSTS-T minus DocT-T). Same highlighting conventions as in Table 5 apply.

| Methods | Train \ Test | FSTS-1.5k F1 | AUC | AFAC F1 | AUC | CertificatePS F1 | AUC | STFD F1 | AUC | FindIt F1 | AUC | Average F1 | AUC |
|---|---|---|---|---|---|---|---|---|---|---|---|---|---|
| **RRU-Net [7]** | Direct | .169 | .692 | .055 | .727 | .211 | .747 | .669 | .957 | .131 | .720 | .247 | .769 |
| | DocT-T | .109 -.060 | .649 -.042 | .079 .024 | .738 .010 | .170 -.040 | .710 -.037 | .652 -.017 | .952 -.005 | .137 .006 | .697 -.023 | .229 -.017 | .749 -.019 |
| | FSTS-T | .320 .151 | .824 .133 | .110 .055 | .788 .060 | .262 .052 | .812 .065 | .712 .043 | .966 .009 | .131 .001 | .733 .013 | .307 .060 | .825 .056 |
| | **Gain** Δ | .211 | .175 | .031 | .050 | .092 | .102 | .060 | .014 | -.006 | .035 | .078 | .075 |
| **DFCN [48]** | Direct | .050 | .693 | .036 | .734 | .062 | .760 | .214 | .898 | .050 | .733 | .082 | .764 |
| | DocT-T | .063 .013 | .669 -.024 | .053 .017 | .752 .018 | .119 .058 | .744 -.016 | .485 .271 | .954 .056 | .115 .064 | .780 .047 | .167 .085 | .780 .016 |
| | FSTS-T | .371 .321 | .867 .173 | .053 .017 | .733 -.001 | .299 .238 | .880 .120 | .576 .362 | .968 .070 | .139 .089 | .770 .037 | .288 .205 | .844 .080 |
| | **Gain** Δ | .309 | .197 | .000 | -.019 | .180 | .136 | .091 | .014 | .024 | -.010 | .121 | .064 |
| **PSCC-Net [28]** | Direct | .175 | .726 | .063 | .684 | .281 | .789 | .351 | .937 | .184 | .764 | .211 | .780 |
| | DocT-T | .203 .028 | .763 .037 | .072 .009 | .683 -.001 | .343 .062 | .820 .031 | .209 -.142 | .904 -.033 | .195 .010 | .780 .016 | .205 -.006 | .790 .010 |
| | FSTS-T | .219 .044 | .790 .064 | .061 -.002 | .684 .000 | .360 .079 | .835 .046 | .354 .003 | .944 .007 | .209 .024 | .780 .016 | .241 .030 | .807 .026 |
| | **Gain** Δ | .016 | .026 | -.011 | .001 | .017 | .015 | .145 | .040 | .014 | .000 | .036 | .017 |
| **MVSS-Net [10]** | Direct | .186 | .623 | .090 | .515 | .326 | .668 | .660 | .898 | .212 | .653 | .295 | .671 |
| | DocT-T | .179 -.007 | .668 .046 | .094 .005 | .641 .126 | .347 .021 | .758 .090 | .629 -.031 | .900 .002 | .271 .059 | .736 .083 | .304 .009 | .740 .069 |
| | FSTS-T | .329 .143 | .746 .123 | .126 .037 | .666 .152 | .421 .095 | .753 .086 | .661 .001 | .906 .006 | .288 .076 | .742 .089 | .365 .070 | .763 .091 |
| | **Gain** Δ | .149 | .077 | .032 | .026 | .074 | -.004 | .032 | .006 | .017 | .006 | .061 | .022 |
| **TruFor [15]** | Direct | .321 | .786 | .146 | .821 | .412 | .862 | .620 | .968 | .273 | .804 | .355 | .848 |
| | DocT-T | .278 -.044 | .750 -.036 | .138 -.008 | .814 -.008 | .375 -.037 | .852 -.010 | .614 -.006 | .959 -.009 | .259 -.015 | .805 .001 | .333 -.022 | .836 -.012 |
| | FSTS-T | .415 .094 | .855 .069 | .191 .045 | .810 -.012 | .417 .005 | .881 .019 | .671 .051 | .973 .005 | .263 -.010 | .823 .019 | .391 .036 | .868 .020 |
| | **Gain** Δ | .137 | .105 | .052 | -.004 | .042 | .029 | .057 | .014 | .004 | .018 | .059 | .032 |
| **DTD [32]** | Direct | .279 | .807 | .012 | .530 | .349 | .825 | .461 | .902 | .145 | .716 | .249 | .756 |
| | DocT-T | .234 -.045 | .761 -.046 | .007 -.005 | .496 -.034 | .234 -.116 | .761 -.064 | .558 .097 | .916 .014 | .102 -.043 | .694 -.022 | .227 -.022 | .726 -.030 |
| | FSTS-T | .291 .012 | .815 .008 | .011 -.001 | .524 -.006 | .353 .004 | .824 -.001 | .600 .139 | .932 .030 | .162 .017 | .734 .017 | .284 .034 | .766 .010 |
| | **Gain** Δ | .057 | .054 | .004 | .028 | .119 | .063 | .042 | .016 | .061 | .039 | .057 | .040 |
| **STFL-Net [45]** | Direct | .205 | .738 | .122 | .799 | .341 | .818 | .683 | .972 | .247 | .808 | .320 | .827 |
| | DocT-T | .226 .021 | .736 -.002 | .120 -.002 | .785 -.014 | .346 .004 | .814 -.004 | .689 .006 | .972 .000 | .248 .001 | .805 -.002 | .326 .006 | .822 -.004 |
| | FSTS-T | .367 .161 | .834 .096 | .132 .010 | .802 .003 | .445 .103 | .866 .048 | .715 .032 | .975 .003 | .307 .060 | .841 .034 | .393 .073 | .864 .037 |
| | **Gain** Δ | .140 | .098 | .012 | .017 | .099 | .052 | .026 | .003 | .059 | .036 | .067 | .041 |

data. However, some methods trained on FSTS-T still exhibit suboptimal performance on specific real-world datasets compared to models trained on other synthetic datasets. For example, PSCC-Net performs poorly on AFAC, and DTD underperforms on STFD. This could be attributed to the models' inherent limitations in extracting discriminative features from low-texture text images, as discussed in the main paper. On the other hand, models trained on synthetic datasets such as TIC13, T-SROIE, and OSTF show better performance on these specific test sets. This is because TIC13, T-SROIE, and OSTF contain some images with textures and content that are more similar to those found in AFAC and STFD, which enables models trained on these datasets to handle these scenarios more effectively.

## C.2  Extended Protocol 3 Evaluation

As shown in Table 6, models trained on real-world data (e.g., STFD) in the "Direct" row exhibit solid performance within their respective dataset. However, their performance suffers significantly when tested on cross-dataset real-world scenarios, showing limited generalization across almost all real-world test sets. This highlights the inability of models trained solely on a specific real-world dataset to generalize effectively to others, as they fail to capture the diversity of tampering distributions in unseen datasets. In fact, when compared with models trained on other real-world datasets, such as CertificatePS (from Protocol 3 in the main paper), the models trained on STFD demonstrate even poorer generalization across cross-dataset tests. This suggests that even though STFD is a real-world dataset, its inability to cover a wider variety of tampered images limits the generalization capability of the trained models.It further emphasizes that training data must incorporate a more diverse set of tampering scenarios to enhance model generalization. Without sufficient variety in the training data, the model is less equipped to generalize well to new, unseen datasets.

Table 7: Pixel-level F1 and AUC performance of T-IFL under the extended Protocol 1, evaluated on three deep generative model forgery datasets: TIC13 [41], T-SROIE [42], and OSTF [33]. Each method includes five rows corresponding to different training–testing settings. The first row reports results for models trained on FSTS-T (**Ours**), followed by four rows trained on existing synthetic datasets (DocT-T, T-SROIE, TIC13, OSTF). Performance gains (**others minus FSTS-T**) are shown as subscripts. Same highlighting conventions as in Table 5 apply.

| Methods | Test / Train | TIC13 F1 | TIC13 AUC | T-SROIE F1 | T-SROIE AUC | OSTF F1 | OSTF AUC | Average F1 | Average AUC |
|---|---|---|---|---|---|---|---|---|---|
| | **FATS-T** | .392 | .774 | .577 | .901 | .379 | .816 | .450 | .831 |
| MVSS-Net [10] | DocT-T | .175 -.217 | .692 -.082 | .479 -.098 | .866 -.035 | .168 -.211 | .709 -.108 | .274 -.176 | .756 -.075 |
| | TIC13 | – | – | .216 -.362 | .862 -.039 | .575 .196 | .928 .111 | .395 -.054 | .895 .065 |
| | T-SROIE | .079 -.314 | .614 -.160 | – | – | .066 -.313 | .589 -.227 | .072 -.377 | .602 -.229 |
| | OSTF | .583 .190 | .842 .068 | .494 -.083 | .894 -.008 | – | – | .538 .089 | .868 .037 |
| | **FATS-T** | .541 | .903 | .776 | .993 | .562 | .939 | .626 | .945 |
| TruFor [15] | DocT-T | .228 -.313 | .864 -.039 | .628 -.148 | .969 -.023 | .245 -.317 | .879 -.060 | .367 -.259 | .904 -.041 |
| | TIC13 | – | – | .179 -.597 | .871 -.122 | .615 .053 | .953 .014 | .397 -.229 | .912 -.033 |
| | T-SROIE | .320 -.221 | .849 -.054 | – | – | .279 -.283 | .849 -.091 | .299 -.327 | .849 -.096 |
| | OSTF | .608 .067 | .851 -.052 | .607 -.168 | .958 -.035 | – | – | .608 -.018 | .905 -.040 |
| | **FATS-T** | .238 | .742 | .428 | .931 | .181 | .775 | .282 | .816 |
| DTD [32] | DocT-T | .047 -.191 | .683 -.059 | .450 .023 | .938 .008 | .118 -.063 | .734 -.041 | .205 -.077 | .785 -.031 |
| | TIC13 | – | – | .192 -.236 | .864 -.067 | .600 .419 | .938 .164 | .396 .114 | .901 .085 |
| | T-SROIE | .032 -.206 | .622 -.120 | – | – | .016 -.166 | .568 -.207 | .024 -.258 | .595 -.221 |
| | OSTF | .543 .305 | .829 .087 | .220 -.208 | .844 -.087 | – | – | .381 .099 | .837 .021 |
| | **FATS-T** | .513 | .913 | .636 | .990 | .491 | .927 | .547 | .943 |
| STFL-Net [45] | DocT-T | .256 -.257 | .856 -.057 | .546 -.089 | .967 -.023 | .267 -.224 | .870 -.057 | .356 -.190 | .898 -.045 |
| | TIC13 | – | – | .234 -.401 | .898 -.092 | .617 .126 | .942 .015 | .425 -.121 | .920 -.023 |
| | T-SROIE | .047 -.466 | .820 -.093 | – | – | .049 -.442 | .834 -.093 | .048 -.498 | .827 -.116 |
| | OSTF | .568 .055 | .883 -.029 | .521 -.115 | .937 -.053 | – | – | .544 -.002 | .910 -.033 |

## C.3 Extended Protocol 4 Evaluation

As shown in Table 6, models pretrained on FSTS-T and then fine-tuned on the STFD dataset consistently outperform their DocT-T counterparts. Specifically, for each tested model, FSTS-T pretraining yields significant gains in both F1 and AUC metrics across the majority of real-world datasets. These improvements emphasize the generalizability of the synthetic data in enhancing the model's ability to perform on unseen, real-world data, particularly when the real-world dataset is limited in size and diversity. In contrast, models pretrained on DocT-T and fine-tuned on STFD show more modest performance, and in some cases (e.g., RRU-Net, PSCC-Net, DTD, TruFor, STFL-Net), they even exhibit negative gains in the average performance when compared to the Direct models trained directly on STFD. This further highlights the advantage of our proposed FSTS-T dataset over conventional synthetic datasets for improving real-world image forgery localization performance.

## C.4 Extended Protocol 1 Evaluation: Deep Generative Models Forgery Datasets

To further assess cross-mechanism generalization, we extend Protocol 1 by evaluating four representative detectors (MVSS-Net, TruFor, DTD, STFL-Net) on three deep generative models forgery datasets (TIC13, T-SROIE, OSTF), generated by GAN- and Diffusion-based pipelines. As shown in Table 7, FSTS-T consistently outperforms DocT-T across all detectors and datasets, with several cases exhibiting over 17% mean F1 improvement. These results indicate that the structured and interpretable real-world tampering distributions modeled by FSTS enable substantially stronger cross-dataset generalization than the rule-based synthetic operations in DocT-T. **An important and non-trivial observation** is that FSTS-trained models generalize well to deep generative model forgery datasets, despite being produced by fundamentally different pipelines. In particular, despite the mechanistic gap between human-edited forgeries and GAN/Diffusion-based generations, FSTS-trained models achieve competitive or even superior performance compared to models trained directly on TIC13, T-SROIE, or OSTF. This demonstrates that modeling real-world tampering distributions endows detectors with strong cross-source and cross-mechanism generalization. We additionally note that models trained on OSTF occasionally achieve stronger results on TIC13 or T-SROIE. This is likely because OSTF includes forgeries produced by more than ten generative models, some of which overlap with or exhibit generation behaviors similar to the models used in these datasets. Such inter-dataset similarity may lead to higher in-domain performance rather than reflecting stronger cross-source generalization.

**These findings further suggest** that the current trend in the community, which places increasing emphasis on the risks of deep generative forgeries and continuously introduces new datasets tailored to specific GAN or Diffusion models, may not provide a fully sustainable path toward robust generalization. Our experiments offer an important complementary insight: the diverse real-world tampering distributions captured by FSTS contribute meaningfully more to generalization than incremental additions of model-specific synthetic data. This highlights the value of enriching real-world tampering distributions at the data level, rather than repeatedly expanding datasets tied to particular generative models.

## D  Limitations

While our approach to modeling synthetic tampering distributions to approximate real-world distributions has demonstrated promising results, there are limitations to the scope of the current model. Specifically, the tampered samples we model are based on a finite set of real-world scenarios. Collecting and analyzing video and historical records for such data is time-consuming and resource-intensive, highlighting the need for more efficient data collection methods. Although we have made efforts to approximate the real-world tampering distribution, there remains a possibility that additional variations in tampering types or methods, which are less represented in our current dataset, could further enhance the model's performance. Expanding the range of tampering behaviors and samples to more comprehensively cover real-world tampering patterns would likely improve the model's generalization capabilities across unseen data.

Table 8: Tampering parameter configurations for the **Copy-move** tampering type, including both *main processing* and *post-processing* operations. The first two columns represent the step index and step name (e.g., Region Sampling, Geometric Transformation, Visual Trace Concealment), which organize related tampering operations for clarity. The third and fourth columns list the specific operation ID and its corresponding description under each step. The remaining columns specify the parameter type, value range, and usage frequency. All processing steps follow a parent-to-child index structure (e.g., 1.1 → 1.2 → 2.1 → 2.2). At each hierarchy level, multiple sub-operations with the same index (e.g., several 2.1 entries) represent mutually exclusive options. In such cases, the frequency values indicate the preferred variants to be selected during synthesis.

| Step | | | Main Processing | Parameter Type | Parameter Value | Frequency |
|---|---|---|---|---|---|---|
| 1 | Region Sampling | 1.1 | Text Region Selection | Region Quantity | 1–12 zones | 100.00% |
| | | 1.2 | Copy Region from Source Image (Within Image) | Text Region | Randomly Select Text Region | 100.00% |
| | | 1.3 | Number of Characters Retained in Source Region | Text Length | 1–20 characters | 100.00% |
| | | 1.4 | Paste Target Region Selection | Target Region | Text Region in Target Image / Copy Area Nearby (9-Grid Positions) | 100.00% |
| 2 | Geometric Transformation | 2.1 | Magic Wand Tool for Text Shape Extraction | Tolerance / Contiguous / Anti-alias | 1-50 / Yes/No / Yes/No | 18.53% |
| | | 2.1 | Adjust channels and levels to remove background and extract text shape | Channel / Input Levels / Output Levels | Red / 130-237 / 0-255 | 23.74% |
| | | 2.2 | Region Scaling | Scaling Factor | Adaptive Scaling to Match Paste Region | 73.50% |
| | | 2.3 | Region Rotation | Rotation Angle | 0°-5° | 13.33% |

| Step | | | Post-processing | Parameter Type | Parameter Value | Frequency |
|---|---|---|---|---|---|---|
| 3 | Visual Trace Concealment | 3.1 | Sharpen | Amount / Radius / Threshold | 100-200% / 1-4 pixels / 7-12 levels | 8.90% |
| | | 3.2 | Blur Filter | Default Parameters | Default Parameters | 5.71% |
| | | 3.2 | Blur More Filter | Default Parameters | Default Parameters | 3.74% |
| | | 3.2 | Mean Filter | Default Parameters | Default Parameters | 5.50% |
| | | 3.2 | Gaussian Blur | Radius | 0.1–3 pixels | 12.70% |
| | | 3.2 | Motion Blur | Angle / Radius | -15°–15° / 1–9 px | 7.10% |
| | | 3.2 | Radial Blur | Method / Quality | Spin/Zoom / Best/Draft/Good | 3.09% |
| | | 3.2 | Smart Blur | Radius / Threshold / Blur Quality / Blur Mode | 0.1–10 pixels / 0.1–10 levels / High/Medium/Low / Edge Preservation/Normal /Stroke Enhancement | 8.78% |
| | | 3.2 | Custom Convolution Filter | Kernel / Scale / Offset | -10–10 / 1–20 / -5–5 | 8.70% |
| | | 3.3 | Color Balance | Tonal Range / Color Sliders | Midtones / -100–100 | 4.18% |
| | | 3.4 | Color Curves | Curve | Raise Highlights /Lower Shadows | 8.53% |

Table 9: Tampering parameter configurations for the **Splicing** tampering type. The tampering process is organized into three steps: Region Sampling, Geometric Transformation, and Visual Trace Concealment. The structural layout and notation follow Table 8.

| Step | | | Main Processing | Parameter Type | Parameter Value | Frequency |
|------|---|---|-----------------|----------------|-----------------|-----------|
| 1 | Region Sampling | 1.1 | Text Region Selection | Region Quantity | 1–12 zones | 100.00% |
| | | 1.2 | Copy Region from Source Image (Cross-Image) | Text Region | Randomly Select Text Region | 100.00% |
| | | 1.3 | Number of Characters Retained in Source Region | Text Length | 1–20 characters | 100.00% |
| | | 1.4 | Paste Target Region Selection | Target Region | Text Region in Target Image | 100.00% |

| Step | | | Main Processing | Parameter Type | Parameter Value | Frequency |
|------|---|---|-----------------|----------------|-----------------|-----------|
| 2 | Geometric Transformation | 2.1 | Magic Wand Tool for Text Shape Extraction | Tolerance | 1-50 | 12.69% |
| | | | | Contiguous | Yes/No | |
| | | | | Anti-alias | Yes/No | |
| | | 2.1 | Adjust channels and levels to remove background and extract text shape | Channel | Red | 17.94% |
| | | | | Input Levels | 130-237 | |
| | | | | Output Levels | 0-255 | |
| | | 2.2 | Region Scaling | Scaling Factor | Adaptive Scaling to Match Paste Region | 78.00% |
| | | 2.3 | Region Rotation | Rotation Angle | 0°-5° | 19.30% |

| Step | | | Post-processing | Parameter Type | Parameter Value | Frequency |
|------|---|---|-----------------|----------------|-----------------|-----------|
| 3 | Visual Trace Concealment | 3.1 | Sharpen | Amount | 100-200% | 10.04% |
| | | | | Radius | 1-4 pixels | |
| | | | | Threshold | 7-12 levels | |
| | | 3.2 | Gaussian Blur | Radius | 0.1–3 pixels | 18.70% |
| | | 3.2 | Lens Blur | Depth Map Mode | None | 7.03% |
| | | | | Invert | Disabled | |
| | | | | Aperture Shape | Hexagon/Heptagon /Octagon/Pentagon /Quadrilateral/Triangle | |
| | | | | Aperture Radius | 0–1 | |
| | | | | Blade Curvature | 0–1 | |
| | | | | Rotation Angle | 0°– 6° | |
| | | | | Brightness | 100% | |
| | | | | Threshold | 0–100% | |
| | | | | Amount | 0–25% | |
| | | | | Distribution | Gaussian/Uniform | |
| | | 3.2 | Motion Blur | Angle | -15°–15° | 5.40% |
| | | | | Radius | 1–9 px | |
| | | 3.2 | Radial Blur | Method | Spin/Zoom | 3.11% |
| | | | | Quality | Best/Draft/Good | |
| | | 3.2 | Smart Blur | Radius | 0.1–10 pixels | 3.90% |
| | | | | Threshold | 0.1–10 levels | |
| | | | | Blur Quality | High/Medium/Low | |
| | | | | Blur Mode | Edge Preservation /Stroke Enhancement/Normal | |
| | | 3.2 | Custom Convolution Filter | Kernel | -10–10 | 2.70% |
| | | | | Scale | 1–20 | |
| | | | | Offset | -5–5 | |
| | | 3.3 | Color Curves | Curve | Raise Highlights /Lower Shadows | 17.45% |
| | | 3.4 | Stroke | Size | 1-5 pixels | 8.75% |
| | | | | Position | Inside/Center/Outside | |
| | | | | Blend Mode | Normal/Multiply | |
| | | | | Opacity | 50%-100% | |
| | | | | Fill Type | color | |
| | | | | Color | RGB(0-255, 0-255, 0-255) | |
| | | 3.5 | Drop Shadow | Blend Mode | Normal/Multiply/Darken | 6.99% |
| | | | | Color | RGB(0-255, 0-255, 0-255) | |
| | | | | Opacity | 5%-23% | |
| | | | | Angle | -30°-30° | |
| | | | | Distance | 1-7 pixels | |
| | | | | Spread | 3%-12% | |
| | | | | Size | 1-17 pixels | |
| | | | | Noise | 1%-10% | |
| | | 3.6 | Hue/Saturation | Hue | -30-–30 | 10.49% |
| | | | | Saturation | -20–20 | |
| | | | | Lightness | -30-–30 | |

Table 10: Tampering parameter configurations for the **Removal** tampering type. The tampering process is organized into three steps: Region Sampling, Content Removal, and Geometric Transformation. The structural layout and notation follow Table 8.

| Step | | | Main Processing | Parameter Type | Parameter Value | Frequency |
|---|---|---|---|---|---|---|
| 1 | Region Sampling | 1.1 | Text Region Selection | Region Quantity | 1–12 zones | 100.00% |
| | | 1.2 | Text Forgery Control | Text Length | 1–20 characters | 100.00% |
| 2 | Content Removal | 2.1 | Content Aware Fill | Iterations | 1-5 times | 55.82% |
| | | 2.1 | Solid Color Fill | Color | RGB(0-255, 0-255, 0-255) | 9.76% |
| | | 2.1 | Pure Background Cloning | Blending Modes | Normal | 11.52% |
| | | 2.1 | Clone Stamp Tool | Mode | Normal | 10.45% |
| | | | | Opacity | 100% | |
| | | | | Flow | 100% | |
| | | 2.1 | Healing Brush Tool | Mode | Normal/Replace | 12.45% |
| | | | | Source | Sampled | |
| 3 | Geometric Transformation | 3.1 | Region Scaling | Scaling Factor | Adaptive to text region ±5% | 88.00% |
| | | 3.2 | Region Rotation | Rotation Angle | -5°-5° | 0.68% |

Table 11: Tampering parameter configurations for the **Insertion** tampering type. The tampering process is organized into four steps: Region Sampling, Text Insertion, Geometric Transformation, and Visual Trace Concealment. The structural layout and notation follow Table 8.

| Step | | | Main Processing | Parameter Type | Parameter Value | Frequency |
|---|---|---|---|---|---|---|
| 1 | Region Sampling | 1.1 | Non-text Region Selection | Region Quantity | 1-12 zones | 100.00% |
| | | 1.2 | Text Forgery Control | Text Length | 1-20 characters | 100.00% |
| 2 | Text Insertion | 2.1 | Font Properties | Fonts | Times New Roman/SimSun /KaiTi/Microsoft YaHei/SimHei | 100.00% |
| | | | | Anti-aliasing | None/Sharp/Crisp /Smooth/Strong | |
| | | 2.2 | Color Adaptation | Color Sampling | Same as the original text color | 86.90% |
| | | 2.2 | Color Selection | Safety Color Generation | Light Background: RGB(0-64, 0-64, 0-64) Dark Background: RGB(192-255, 192-255, 192-255) | 13.10% |
| 3 | Geometric Transformation | 3.1 | Region Scaling | Scaling Factor | Adaptive to text region ±5% | 77.00% |
| | | 3.2 | Region Rotation | Rotation Angle | -5°-5° | 12.03% |

| Step | | | Post-processing | Parameter Type | Parameter Value | Frequency |
|---|---|---|---|---|---|---|
| 4 | Visual Trace Concealment | 4.1 | Sharpen | Iterations | 1-5 times | 12.73% |
| | | | | Strength | 400-500% | |
| | | | | Radius | 50-60 pixels | |
| | | | | Threshold | 2-3 levels | |
| | | 4.2 | Gaussian Blur | Blur Radius | 0.5-1.2 pixels | 16.80% |
| | | 4.3 | Outer Glow Effect | Color | RGB(83,79,79) | 7.51% |
| | | | | Opacity | 17% | |
| | | | | Noise | 35-45% | |
| | | | | Spread | 5-8px | |
| | | 4.4 | Noise | Amount | 0.10%-35% | 16.54% |
| | | | | Distribution | Gaussian/Uniform | |
| | | | | Monochromatic | Yes/No | |
| | | 4.5 | Stroke | Size | 1-5 pixels | 15.33% |
| | | | | Position | Inside/Center/Outside | |
| | | | | Blend Mode | Normal/Multiply | |
| | | | | Color | RGB(0-255, 0-255, 0-255) | |
| | | 4.6 | Drop Shadow | Blend Mode | Normal/Multiply/Darken | 5.26% |
| | | | | Color | RGB(0-255, 0-255, 0-255) | |
| | | | | Opacity | 5%-23% | |
| | | | | Angle | -30°-30° | |
| | | | | Distance | 1-7 pixels | |
| | | | | Spread | 3%-12% | |
| | | | | Size | 1-17 pixels | |
| | | | | Noise | 1%-10% | |

Table 12: Tampering parameter configurations for the **Replacement** tampering type. The tampering process is organized into five steps: Region Sampling, Content Removal, Text Insertion, Geometric Transformation, and Visual Trace Concealment. The structural layout and annotations follow Table 8.

| | Step | | Main Processing | Parameter Type | Parameter Value | Frequency |
|---|---|---|---|---|---|---|
| 1 | Region Sampling | 1.1 | Text Region Selection | Region Quantity | 1–12 zones | 100.00% |
| | | 1.2 | Text Forgery Control | Text Length | 1–20 characters | 100.00% |
| 2 | Content Removal | 2.1 | Content Aware Fill | Iterations | 1-5 times | 61.70% |
| | | 2.1 | Solid Color Fill | Color | RGB(0-255, 0-255, 0-255) | 9.60% |
| | | 2.1 | Pure Background Cloning | Blending Modes | Normal | 9.50% |
| | | 2.1 | Clone Stamp Tool | Mode | Normal | 10.40% |
| | | | | Opacity | 100% | |
| | | | | Flow | 100% | |
| | | 2.1 | Healing Brush Tool | Mode | Normal/Replace | 8.80% |
| | | | | Source | Sampled | |
| 3 | Text Insertion | 3.1 | Font Properties | Fonts | Times New Roman/SimSun /KaiTi/Microsoft YaHei/SimHei | 100.00% |
| | | | | Anti-aliasing | None/Sharp/Crisp/Smooth/Strong | 100.00% |
| | | 3.3 | Color Adaptation | Color Sampling | Same as the original text color | 88.40% |
| | | 3.4 | Color Selection | Safety Color Generation | Light Background: RGB(0-64, 0-64, 0-64) Dark Background: RGB(192-255, 192-255, 192-255) | 11.60% |
| 4 | Geometric Transformation | 4.1 | Region Scaling | Scaling Factor | Adaptive to text region ±5% | 43.50% |
| | | 4.1 | Region Rotation | Rotation Angle | -5°-5° | 33.33% |

| | Step | | Post-processing | Parameter Type | Parameter Value | Frequency |
|---|---|---|---|---|---|---|
| 5 | Visual Trace Concealment | 5.1 | Sharpen | Iterations | 1-5 times | 12.69% |
| | | | | Strength | 400-500% | |
| | | | | Radius | 50-60 pixels | |
| | | | | Threshold | 2-3 levels | |
| | | 5.2 | Gaussian Blur | Blur Radius | 0.5-1.2 pixels | 11.91% |
| | | 5.2 | Surface Blur | Radius | 1-15 pixels | 7.60% |
| | | | | Threshold | 5-25 levels | |
| | | 5.2 | Motion Blur | Angle | -30°-30° | 7.63% |
| | | | | Distance | 1-20 pixels | |
| | | 5.3 | Outer Glow Effect | Color | RGB(83,79,79) | 13.68% |
| | | | | opacity | 17% | |
| | | | | Noise | 35-45% | |
| | | 5.4 | Noise | Amount | 0.10%-35% | 10.47% |
| | | | | Distribution | Gaussian/Uniform | |
| | | | | Monochromatic | Yes/No | |
| | | 5.5 | Stroke | Size | 1-5 pixels | 10.20% |
| | | | | Position | Inside/Center/Outside | |
| | | | | Blend Mode | Normal/Multiply | |
| | | | | Opacity | 50%-100% | |
| | | | | Fill Type | color | |
| | | | | Color | RGB(0-255, 0-255, 0-255) | |
| | | 5.6 | Drop Shadow | Blend Mode | Normal/Multiply/Darken | 8.81% |
| | | | | Color | RGB(0-255, 0-255, 0-255) | |
| | | | | Opacity | 5%-23% | |
| | | | | Angle | -30°-30° | |
| | | | | Distance | 1-7 pixels | |
| | | | | Spread | 3%-12% | |
| | | | | Size | 1-17 pixels | |
| | | | | Noise | 1%-10% | |

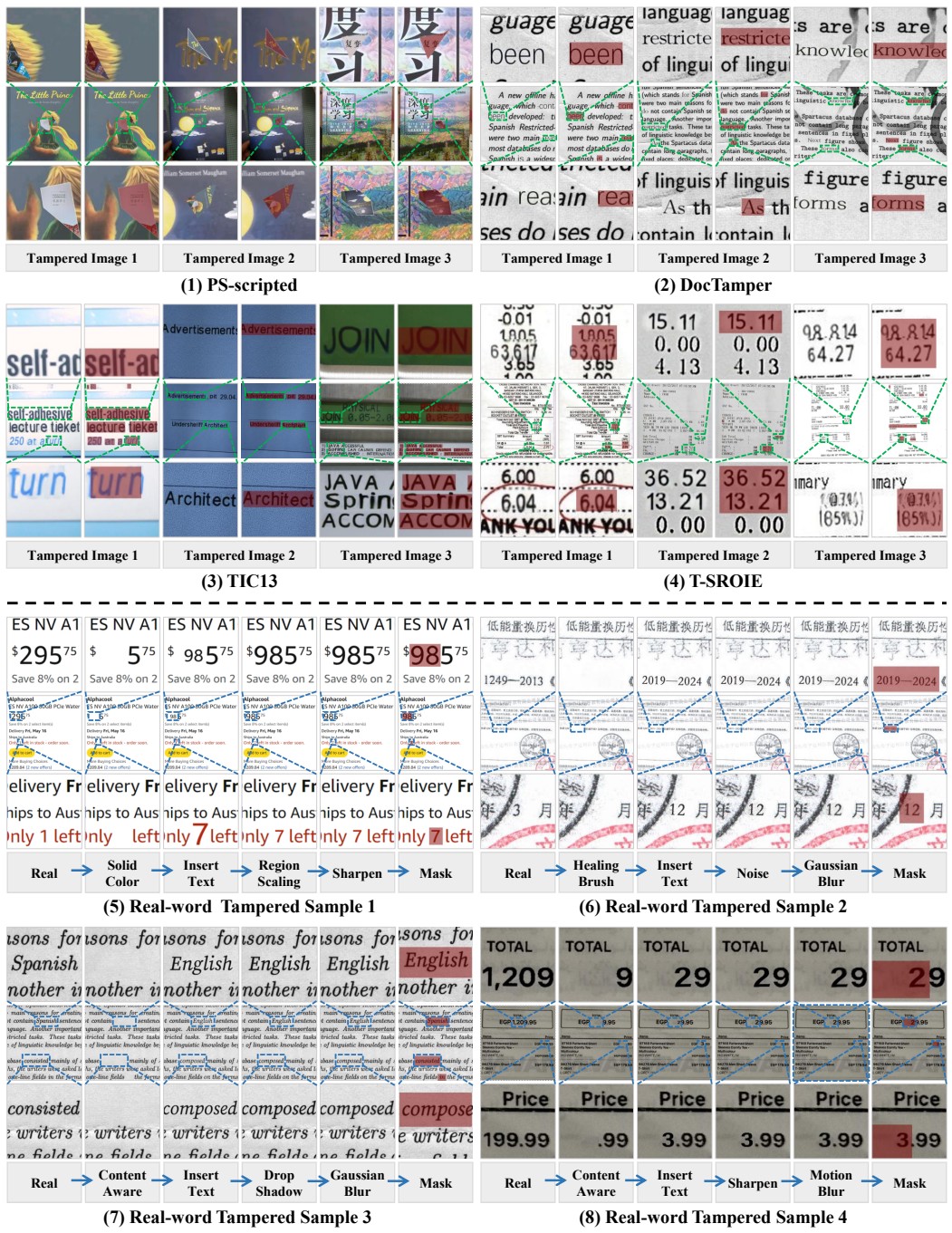

Figure 5: Comparison of tampering operation diversity between existing synthetic datasets and real-world forgery samples. (1)–(4) show examples from four synthetic datasets: PS-scripted [48], DocTamper [32], TIC13 [41], and T-SROIE [42]. In each sample, the forged region is highlighted in red. PS-scripted uses real-world tampering parameters but randomly assigns tampering targets, lacking representative coverage of tampering types. The others are generated using deep generative methods, which often apply similar operations and parameters across samples, reflecting limited diversity in invisible distributions. In contrast, (5)–(8) visualize four replacement samples collected from real-world tampered data. Each case reflects a distinct combination of tampering operation-parameters (e.g., region sampling, insertion, shadow, blur), illustrating the diversity and complexity inherent in real-world tampering behaviors. This comparison highlights the importance of modeling invisible parameter distributions to improve the diversity and realism of synthetic data.

