# OpenReview forum: "Toward Real-world Text Image Forgery Localization: Structured and Interpretable Data Synthesis"
_NeurIPS.cc/2025/Datasets_and_Benchmarks_Track — NeurIPS 2025 Datasets and Benchmarks Track poster_

### Official Review · Reviewer_asrA · 2025-06-24

**Rating:** 6
**Confidence:** 5

**Summary:**

- This paper addresses the critical gap between real-world and synthetic text image forgery localization (T-IFL) by proposing a structured and interpretable dataset synthesis framework called Fourier Series-based Tampering Synthesis (FSTS).

- Unlike prior datasets that emphasize visible attributes such as scene variety, text language, or dataset scale, FSTS focuses on modeling the invisible distribution of tampering parameters, including tampering types, operation sequences, and post-processing effects commonly observed in real-world forgery behaviors.

- To support this, the authors collect 16,750 real-world tampering instances from 67 human tamperers across five representative manipulation types, using a structured pipeline that records editing traces via multi-format logs, enabling the extraction of fine-grained tampering parameters. These parameters are then modeled hierarchically at both individual and population levels using a compact combination of basis configurations and weights, inspired by Fourier series. By sampling from the learned distribution, FSTS synthesizes diverse and realistic tampered text images that better reflect authentic forgery traces.

- Extensive experiments under four evaluation protocols show that models trained on FSTS data achieve significantly improved generalization to real-world datasets, highlighting the value of this benchmark for T-IFL research.

**Additional Feedback:**

- Would the authors consider releasing the synthesis code in the future? Beyond improving reproducibility, such a release may also inspire extensions of the framework to other domains, such as natural image forgery localization.

**Dataset Code Accessibility:**

Yes

**Dataset Code Comments:**

Well-released on the Kaggle platform.

**Ethical Considerations:**

No, there are no or only very minor ethics concerns

**Final Justification:**

I have carefully reviewed the manuscript and the authors’ rebuttal. I find that this work makes a meaningful and well-executed contribution to the dataset track.

The proposed FSTS framework offers a structured and interpretable approach for bridging the gap between synthetic and real-world text image forgeries by modeling the invisible distribution of tampering parameters, resulting in a large-scale, diverse benchmark. Through structured collection of fine-grained real-world tampering logs and hierarchical modeling, the authors produce realistic synthetic data that better reflects authentic forgery traces.

By elaborating on the tampering process, validating dataset comprehensiveness, fixing presentation issues, and securing data access, the authors’ rebuttal effectively alleviated my earlier concerns. These refinements significantly boost the study’s clarity, replicability, and real-world applicability.

The paper presents a valuable resource that is likely to support future research in Text Image Forgery Localization and related forensic domains. I support acceptance.

**Limitations Weaknesses:**

While the proposed FSTS framework is well-motivated and demonstrates strong effectiveness, there are a few aspects that could be further improved or refined in future iterations:

- Minor typo in line 6 of the abstract (“TFSTS” instead of “FSTS”), does not affect clarity, but authors are encouraged to proofread for consistency.

- In Section B.2 of the supplementary material (line 90), the authors are encouraged to add a brief textual description of the tampering process illustrated in Figure 1(II), especially the implementation details of the five representative tampering types. Although the figure caption provides some explanation, an explicit mention in the main paper could help readers from other domains better understand how text image tampering is performed in practice.

**Strengths Contributions:**

The paper addresses a core limitation in text image forgery localization (T-IFL) by shifting the focus from visible dataset attributes (e.g., scene, language, quantity) to the invisible distribution of tampering behaviors, which is more reflective of real-world forgery complexity.

- The authors design a structured pipeline to collect a substantial number of real-world tampering instances via multi-format logging (video, PSD, editing logs), enabling fine-grained parameter extraction. Based on this data, they propose a novel and interpretable synthesis framework, FSTS, which hierarchically models tampering parameters at both individual and population levels using a basis-and-weight formulation inspired by Fourier series. The theoretical formulation is well grounded, and the modeling analysis is conceptually clear and empirically validated.

- The authors release a large-scale tampered text image dataset comprising 294,694 samples across multiple variants (FSTS-T, FSTS-S, FSTS-2k, and FSTS-v2), covering both synthetic and real-world sources. According to the authors, this is currently the largest dataset of its kind. Preliminary experiments on the released dataset suggest that it is effective in improving model performance on real-world T-IFL benchmarks.

- Overall, the paper is well-written and clearly presented, offering a well-structured and timely contribution with practical relevance and potential impact on future work in realistic dataset construction and tampering analysis.

---

> ### Author Rebuttal · Authors · 2025-07-31
>
> We thank the reviewer for the constructive comments. We provide our responses as follows:
>
> **Q1: Minor typo in line 6 of the abstract.**
>
> A1: We thank the reviewer for pointing out the typo in line 6 of the abstract. We will correct “TFSTS” to “FSTS” in the revised version for consistency.
>
> **Q2: In Section B.2 of the supplementary material (line 90), the authors are encouraged to add a brief textual description of the tampering process illustrated in Figure 1(II), especially the implementation details of the five representative tampering types. Although the figure caption provides some explanation, an explicit mention in the main paper could help readers from other domains better understand how text image tampering is performed in practice.**
>
> A2: We thank the reviewer for the helpful suggestion. While the caption of Fig. 1 in the supplementary material already provides a high-level summary, we agree that a concise textual description in the main text or supplementary material could improve clarity, especially for readers from other domains. We have now included a brief explanation of the main processing and post-processing operations for the five representative tampering types:
>
> - **Copy-move:** As shown in Fig. 1(II)(1), a region containing the digit “4” is first selected (*1. Region Sampling (Copy)*) from the target image *$I^o$*, and then pasted to a nearby location (*1. Region Sampling (Paste)*). The copied region is geometrically transformed to fit the destination region and then refined with visual trace concealment to hide tampering artifacts (*2. Geometric Transformation, 3. Visual Trace Concealment*). The resulting tampered image *$I^s$* contains an additional “4”, and the tampered area is highlighted in light blue in the ground truth mask *$M$*.
>
> - **Splicing:** As shown in Fig. 1(II)(2), a region containing the digit “2025” is first selected (*1. Region Sampling (Copy)*) from the source image *$I^{o'}$*, and then pasted onto the region containing “2016” in the target image *$I^o$* (*1. Region Sampling (Paste)*). The copied region is geometrically transformed to fit the destination and further processed with visual trace concealment to hide tampering artifacts (*2. Geometric Transformation, 3. Visual Trace Concealment*). The resulting tampered image *$I^s$* shows the digit “2016” replaced by “2025”, and the tampered area is highlighted in green in the ground truth mask *$M$*.
>
> - **Removal:** As shown in Fig. 1(II)(3), a region containing the “certification mark” is first selected (*1. Region Sampling*) from the target image *$I^o$*, and then the selected content is erased (*2. Content Removal*). The resulting tampered image *$I^s$* no longer contains the certification mark, and the tampered area is highlighted in dark blue in the ground truth mask *$M$*.
>
> - **Insertion:** As shown in Fig. 1(II)(4), a blank region is first selected (*1. Region Sampling*) in the image *$I^o$*, where new text is inserted (*2. Text Insertion*), such as “5964.7M Views”. The inserted text is geometrically transformed to fit the destination and further processed with visual trace concealment to hide tampering artifacts (*3. Geometric Transformation, 4. Visual Trace Concealment*). The resulting tampered image *$I^s$* contains the newly added text, and the tampered region is highlighted in yellow in the ground truth mask *$M$*.
>
> - **Replacement:** As shown in Fig. 1(II)(5), a region containing the text “AMERICA” is first selected (*1. Region Sampling*) from the target image *$I^o$*, and its content is erased (*2. Content Removal*). New text, “CHINA!!!”, is then inserted into the same location (*3. Text Insertion*), geometrically transformed to fit the destination, and further processed with visual trace concealment (*4. Geometric Transformation, 5. Visual Trace Concealment*). The resulting tampered image *$I^s$* shows that the original text “AMERICA” has been replaced with “CHINA!!!”, and the tampered region is highlighted in red in the ground truth mask *$M$*.
>
> **Q3: Would the authors consider releasing the synthesis code in the future? Beyond improving reproducibility, such a release may also inspire extensions of the framework to other domains, such as natural image forgery localization.**
>
> A3: We thank the reviewer for the suggestion. We recognize the importance of reproducibility and the potential value of releasing the synthesis code. We are currently evaluating how to best organize and document the codebase, and we aim to make it publicly available in the near future.

---

> > ### Comment · Reviewer_asrA · 2025-08-05
> >
> > I have carefully reviewed the manuscript and the authors’ rebuttal. I find that this work makes a meaningful and well-executed contribution to the dataset track.
> >
> > The proposed FSTS framework offers a structured and interpretable approach for bridging the gap between synthetic and real-world text image forgeries by modeling the invisible distribution of tampering parameters, resulting in a large-scale, diverse benchmark. Through structured collection of fine-grained real-world tampering logs and hierarchical modeling, the authors produce realistic synthetic data that better reflects authentic forgery traces.
> >
> > By elaborating on the tampering process, validating dataset comprehensiveness, fixing presentation issues, and securing data access, the authors’ rebuttal effectively alleviated my earlier concerns. These refinements significantly boost the study’s clarity, replicability, and real-world applicability.
> >
> > The paper presents a valuable resource that is likely to support future research in Text Image Forgery Localization and related forensic domains. Thus, I raised my score from 5 to 6.

---

### Official Review · Reviewer_aQSz · 2025-06-24

**Rating:** 6
**Confidence:** 5

**Summary:**

This paper proposes FSTS - a novel, structured and interpretable synthesis framework for tampered text images based on hierarchical Fourier series modeling of real tampering behaviors. With FSTS, large-scale, human-annotated dataset of real-world tampering instances is collected by recording editing traces in multiple formats (e.g., video, PSD files, and logs). Inspired by the concept of Fourier series decomposition, the tampering distributions are modeled at both the individual and population levels, enabling the generation of synthetic data that are better consistent with real-world forgery patterns. Experimental results under multiple protocols show that the models trained with FSTS-generated data could achieve superior generalization on real-world text-image forgery localization tasks. The proposed data synthesis method is well-motivated, thoroughly evaluated, and demonstrates clear benefits for improving model generalization to real-world tampered text images.

**Additional Feedback:**

1. For the five defined tampering types, it should be clarified that whether the participants were instructed to follow specific editing procedures, or they were free to perform the tampering operations in their own way within each category. Clarifying this aspect would help readers to better understand how much control was applied during data collection, and how it may affect the diversity and realism of the resulting parameter distribution.
2. I fully recognize the efforts involved in collecting and organizing real-world tampering parameters, which is often overlooked in most existing synthetic forgery datasets. Although the paper clearly presents the methodology, it would be valuable to know whether the authors are willing to release additional synthesis-related resources, such as generation scripts or sample videos of the editing process. These materials could provide more intuitive insights and help to proceed the further developments of text image forgery localization.

**Dataset Code Accessibility:**

Yes

**Ethical Considerations:**

No, there are no or only very minor ethics concerns

**Final Justification:**

I have read the author rebuttal and considered all raised points. All my concerns about the notation consistency, the protocol details to perform the tampering operations. Based on the response, I think this work is ready for acceptance.

**Limitations Weaknesses:**

In Appendix Table 3, the STFD column is highlighted in the table body, probably to indicate its role in Protocol 3, similar to Table 3 in the main paper. Corresponding to this, the word “STFD” should also be highlighted in the caption of Appendix Table 3. For consistency and clarity, it is helpful to also clarify the reason for highlighting STFD in the caption.

**Strengths Contributions:**

1. Novelty and Contribution: This paper proposes a hierarchical Fourier series-inspired framework to model real-world tampering parameters, which is used to synthesize text-image forgeries grounded in observed behaviors. The work is solid and of certain interests to the forensic community.
2. Significance and Impact: By generating realistic and behavior-driven tampered samples, FSTS bridges the gap between synthetic data and real-world attack strategies, making it quite desirable for advancing practical progress in text image forgery localization.
3. Empirical Validation: The experimental setting is reasonable and the experiment is sufficient. Through extensive experiments on four evaluation protocols, the paper demonstrates that the models trained on FSTS-synthesized data consistently outperform those trained on other existing synthetic datasets across diverse real-world benchmarks, indicating the effectiveness of the proposed method.
4. Clarity of Key Ideas: The introduction of the pipeline, from dataset collection to Fourier-based distribution modeling and sample generation, is well-motivated and logically presented, making the core concepts and their rationale easy to understand and reproduce.

---

> ### Author Rebuttal · Authors · 2025-07-31
>
> We thank the reviewer for the constructive comments. We provide our responses as follows:
>
> **Q1: In Appendix Table 3, the STFD column is highlighted in the table body, probably to indicate its role in Protocol 3, similar to Table 3 in the main paper. Corresponding to this, the word “STFD” should also be highlighted in the caption of Appendix Table 3. For consistency and clarity, it is helpful to also clarify the reason for highlighting STFD in the caption.**
>
> A1: We thank the reviewer for pointing this out. We will revise the caption of Appendix Table 3 to explicitly clarify that the STFD column is highlighted to indicate its role as the real-world test domain in Protocol 3, consistent with Table 3 in the main paper.
>
> **Q2: For the five defined tampering types, it should be clarified that whether the participants were instructed to follow specific editing procedures, or they were free to perform the tampering operations in their own way within each category. Clarifying this aspect would help readers to better understand how much control was applied during data collection, and how it may affect the diversity and realism of the resulting parameter distribution.**
>
> A2: We thank the reviewer for this insightful comment. In our data collection pipeline, participants were **not provided with fixed editing procedures**, but were instead given the **basic definitions of the five tampering types** (i.e., copy-move, splicing, removal, insertion, replacement). Within each category, they were encouraged to **freely design the tampering process** (including main processings and post-processing techniques) based on the specific context of the image and forgery intent. This approach was intended to maximize diversity and realism in the collected parameter distributions, while maintaining consistency in manipulation type annotation.
>
> **Q3: I fully recognize the efforts involved in collecting and organizing real-world tampering parameters, which is often overlooked in most existing synthetic forgery datasets. Although the paper clearly presents the methodology, it would be valuable to know whether the authors are willing to release additional synthesis-related resources, such as generation scripts or sample videos of the editing process. These materials could provide more intuitive insights and help to proceed the further developments of text image forgery localization.**
>
> A3: We sincerely thank the reviewer for recognizing our efforts in collecting and organizing real-world tampering parameters. We agree that synthesis-related resources such as generation scripts and editing videos can offer valuable insights for the community.
>
> To facilitate understanding and reproducibility, we have already included multi-format logs in the supplementary materials, including video recordings, Photoshop PSD files, and structured parameter logs for each tampering instance.
>
> Due to the NeurIPS rebuttal policy, we are unable to release additional examples or code during the rebuttal phase. However, we are fully committed to releasing more comprehensive resources in the final version, such as generation scripts, additional illustrative video samples, and documentation that facilitates understanding and reproduction of the synthesis pipeline, as long as the conference policy permits. We believe these materials will further support future research in text image forgery localization.

---

> > ### Comment · Reviewer_aQSz · 2025-08-04
> > **Comments to the Rebuttal**
> >
> > All my concerns were properly addressed. Thanks.

---

> > > ### Author Response · Authors · 2025-08-04
> > >
> > > We sincerely thank the reviewer for carefully reading our rebuttal and confirming that the concerns have been addressed. We greatly appreciate your constructive feedback and support during the review process.

---

### Official Review · Reviewer_k4rU · 2025-07-02

**Rating:** 4
**Confidence:** 5

**Summary:**

This paper proposed Fourier Series-based Tampering Synthesis (FSTS), a structured and interpretable framework for synthesizing tampered text images. Experiments on the several datasets show the effectiveness of this benchmark.

**Dataset Code Accessibility:**

Partly

**Dataset Code Comments:**

From the Dataset Reviewer Report, we can see that there are some issues with the availability of the Data Files, and two files are inaccessible.  Moreover, the description of Core Metadata Presence is missing.

**Ethical Considerations:**

No, there are no or only very minor ethics concerns

**Final Justification:**

The author has resolved all my concerns, so I choose to keep my decision.

**Limitations Weaknesses:**

- Frequency analysis has a wide range of applications in both forged content generation and detection. The authors should add a discussion of these works in the related work section, such as [1], [2], and [3].

[1] Frequency-aware GAN for Adversarial Manipulation Generation. ICCV 2023.

[2] Exploring Frequency Adversarial Attacks for Face Forgery Detection. CVPR 2022.

[3] Frequency-Aware Spatiotemporal Transformers for Video Inpainting Detection. ICCV 2021.
- Please unify the format of references. At least ensure that the citation formats of conferences and journals are consistent.
- Based on the experimental results, the author analyzed the reasons why some methods failed, but it would be more intuitive if some test failure examples could be provided.
- Please standardize the capitalization of English letters in the references. Many abbreviations of proper nouns are incorrect, such as a patch-based cnn (a patch-based CNN).

**Strengths Contributions:**

- The motivation of this paper is clear and the authors chose a straightforward but effective method to achieve the goal.
- The charts related to the experiments in the paper are relatively clear, and the organization of the charts is logical.
- Introducing spectral analysis into structured and interpretable data analysis is very interesting for Text Image Forgery Localization.

---

> ### Author Rebuttal · Authors · 2025-07-31
>
> We thank the reviewer for the constructive comments. We provide our responses as follows:
>
> **Q1: Suggestion to discuss frequency-based methods in generation and detection tasks (e.g., [1], [2], [3]).**
>
> A1: We thank the reviewer for the helpful suggestion. We agree that frequency analysis plays an important role in both forgery generation and detection. We will include a brief discussion of the suggested works [1–3] in the Related Work section of the revised version.
>
> **Q2&Q4: Suggestions to unify reference formatting and standardize capitalization of abbreviations.**
>
> A2&A4: We thank the reviewer for pointing out the formatting issues in the reference list.
> We will (1) unify the citation formats of conference and journal papers to ensure consistency, and (2) correct all improper capitalizations of English abbreviations (e.g., CNN) in the revised version.
>
> **Q3: Suggestion to provide representative test failure cases for more intuitive analysis.**
>
> A3: Due to this year’s NeurIPS rebuttal policy, we are not allowed to include any visualizations or external links during the rebuttal stage. However, we agree that including representative failure cases would improve the clarity of our analysis. We will incorporate corresponding examples and discussion in the revised version.
>
> **Q5: Clarification on minor data files accessibility warnings and autogenerated metadata description.**
>
> A5: We thank the reviewer for pointing out these concerns.
>
> (1) Regarding the file accessibility issue, we have thoroughly re-checked the dataset on Kaggle and confirmed that all files are currently accessible and downloadable without restriction. This may have been caused by a temporary versioning or permission issue during the review period.
>
> (2) Regarding the missing Core Metadata description, we note that Kaggle automatically generates a Croissant-compatible JSON metadata file, but this process does not cover all required OpenReview fields, such as "description".
>
> (3) We had already provided a detailed FSTS_Readme.md file in the dataset before submission, and also included it in the supplementary material. This file documents the overall dataset structure and all sub-dataset configurations. We will refine the metadata file in the revised version, ensuring full structural completeness and compatibility.

---

> > ### Comment · Reviewer_k4rU · 2025-08-07
> > **Response to the Rebuttal**
> >
> > The author has resolved all my concerns, so I choose to keep my decision. I hope the authors can provide a brief discussion of the suggested works and  representative test failure cases in the revised version.

---

> > > ### Author Response · Authors · 2025-08-08
> > >
> > > Thank you for your thoughtful review and constructive feedback throughout the process. We truly appreciate the time and effort you have dedicated to evaluating our work, and your comments have been valuable in helping us improve the paper.

---

> ### Comment · Area_Chair_nxDW · 2025-08-06
>
> Dear k4rU,
>
> Could you take a look over rebuttal response and leave your feedback?
>
> Also, you need to flag mandatory acknowledgement button after feedbacks.
>
> AC

---

> ### Author Response · Authors · 2025-08-07
>
> Thank you again for your thoughtful review. If you have any further questions or would like to see additional experiments, please let us know as soon as possible so that we have adequate time to prepare and share the results before the discussion period ends.

---

### Official Review · Reviewer_1iDP · 2025-07-03

**Rating:** 4
**Confidence:** 4

**Summary:**

The paper introduces Fourier Series-based Tampering Synthesis (FSTS), a framework for generating realistic tampered text images by modeling the "invisible distribution" of real-world tampering parameters.

**Dataset Code Accessibility:**

Yes

**Ethical Considerations:**

No, there are no or only very minor ethics concerns

**Final Justification:**

The rebuttal has addressed all my concerns. This dataset will help the IFDL field much and I hope that it would be opened for future research.

**Limitations Weaknesses:**

* This paper just focuses on five text-specific manipulations (e.g., copy-move, splicing) but omits broader edits (e.g., style transfer, semantic manipulations) which may hinder the diversity of real-world scenarios.
* This paper does not provide clear resource requirements for training models on FSTS-generated data, limiting accessibility for low-resource researchers.
* This paper lacks some state-of-the-art methods in IMDL such as TruFor [1], or HiFi [2]. The authors are suggested to include this in the rebuttal.

References \
[1] Guillaro, Fabrizio, et al. "Trufor: Leveraging all-round clues for trustworthy image forgery detection and localization." Proceedings of the IEEE/CVF conference on computer vision and pattern recognition. 2023.
[2] Guo, Xiao, et al. "Hierarchical fine-grained image forgery detection and localization." Proceedings of the IEEE/CVF Conference on Computer Vision and Pattern Recognition. 2023.

**Strengths Contributions:**

* This paper first introduce a method to model "invisible" tampering parameters (e.g., operation sequences, post-processing) beyond visible attributes like scene diversity. FSTS’s interpretability via basis functions aligns with real-world forgery patterns, validated by human behavioral analysis.
* The results of testing four protocols, including synthetic-to-real and pretraining-finetuning scenarios, showing consistent gains (e.g., +21% F1 for DTD on real data).
* This paper introduce a way address data scarcity and privacy via synthetic data, with safeguards for PII removal.

---

> ### Author Rebuttal · Authors · 2025-07-31
>
> We thank the reviewer for the constructive comments. We provide our responses as follows:
>
> **Q1: This paper just focuses on five text-specific manipulations (e.g., copy-move, splicing) but omits broader edits (e.g., style transfer, semantic manipulations) which may hinder the diversity of real-world scenarios.**
>
> A1: We thank the reviewer for their suggestion regarding the diversity of tampering types. We would like to clarify the following:
>
> (1) This work focuses on the task of Text Image Forgery Localization (T-IFL), which aims to localize tampered regions in common real-world text images. The five tampering types we consider (copy-move, splicing, removal, insertion, replacement) have been **widely adopted in previous works such as STFD and CertificatePS (see Table 1 in the Appendix).** To the best of our knowledge, these five types of manipulations are the most recognized and complete attacks to evaluate the performance of T-IFL algorithms in image forgery forensic community.
>
> (2) The “style transfer” and “semantic manipulations” mentioned by the reviewer are **primarily applied in natural image generation or style transfer tasks, where the goal is to modify global appearance or semantics.** These types of global manipulations are rarely observed in real-world T-IFL, which typically involves localized edits to key textual regions.
>
> (3) **Our dataset covers a wide range of real-world scenarios, including receipts, certificates, academic reports, and screenshots, ensuring scene-level diversity and practical relevance.** Detailed information can be found in Appendix B.3 Dataset Variants (Lines 91–112).
>
> We hope this clarifies our focus on representative text-specific manipulations and the real-world diversity covered in our dataset.
>
> **Q2: This paper does not provide clear resource requirements for training models on FSTS-generated data, limiting accessibility for low-resource researchers.**
>
> A2: We thank the reviewer for the concern regarding resource requirements for training models on FSTS-generated data. We clarify the following:
>
> (1) This work proposes a data synthesis framework and does not introduce any new model architecture or training pipeline, nor does it impose additional computational burdens. All experiments are based on public SOTA methods (e.g., MVSS-Net, DTD), strictly following their official implementations and default hyperparameters.
>
> (2) Our experimental setting follows the configuration used in DocTamper, with an input image size of 512×512 (Appendix, Lines 13–15), and models are trained on 50,000 images for 25 epochs (Section 4.1, Lines 256–264).
> To accelerate training, we conducted all experiments using an NVIDIA A100 (80GB) GPU. Only minor adjustments to the batch size were made to accommodate memory constraints, without altering any other settings. **Importantly, the training process is not dependent on this specific hardware, as all SOTA methods can also be trained successfully on other widely available GPUs such as the RTX 3090 (24GB) without any code or configuration change. This has been consistently validated in prior studies using the same public baselines.**
>
> (3) The resource consumption (e.g., GPU memory and training time) of six SOTA methods under Protocols 1 or 2 is summarized below based on the experiments reported in the main paper, where models are trained on either FSTS-T or DocT-T depending on the protocol.
>
> | Method    | Batchsize | GPU memory (GB) | Training Time (h) |
> |-----------|-----------|------------------|--------------------|
> | RRU-Net   | 36        | 75.4            | 14                 |
> | DFCN      | 72        | 63.6            | 11                 |
> | PSCC-Net  | 8         | 77.2            | 30                 |
> | MVSS-Net  | 48        | 58.0            | 7                  |
> | DTD       | 24        | 72.4            | 15                 |
> | STFL-Net  | 16        | 77.1            | 30                 |
>
>
> These results demonstrate that our experiments are feasible for researchers with limited computational resources. We will clarify this point more explicitly in the revised version.
>
> **Q3: This paper lacks some state-of-the-art methods in IMDL such as TruFor [1], or HiFi [2]. The authors are suggested to include this in the rebuttal.**
>
> A3: We thank the reviewer for pointing out the absence of some recent IFL methods. We clarify the following:
>
> (1) **Baseline selection strategy.** Instead of proposing a new T-IFL model, our work aims to validate the generalization and effectiveness of the proposed FSTS data synthesis framework. Therefore, we prioritize **fully open-sourced and widely adopted SOTA methods** to ensure **fairness and reproducibility** in all experiments.
>
> - TruFor (CVPR 2023) is indeed a representative method in IFL. **However, until the time of our paper submission, only the testing code of TruFor was available, and its training pipeline (involving noise modeling, localization, and detection stages) was not yet released.** To avoid implementation bias, we excluded it initially. Fortunately, we note that the authors publicly released the training code approximately 2–3 months after our submission, and we have since included it in our supplementary experiments.
>
> - HiFi-Net (CVPR 2023) was originally developed for **multi-class natural image forgery detection and localization, requiring both binary authenticity labels (real/fake) and manipulation-type annotations (e.g., GAN, Diffusion) during training.** Since such labels are not applicable in T-IFL, we adapted HiFi-Net to binary classification with “real vs. fake” supervision in our supplementary experiments.
>
> (2) To further verify the generalization ability of our proposed FSTS dataset, we also include two recently emerged models after our submission, namely SparseViT [R1] and Mesorch [R2] from AAAI 2025, in our evaluation.
>
> - [R1] Can we get rid of handcrafted feature extractors? sparsevit: Nonsemantics-centered, parameter-efficient image manipulation localization through spare-coding transformer.
>
> - [R2] Mesoscopic Insights: Orchestrating Multi-scale & Hybrid Architecture for Image Manipulation Localization.
>
> (3) Due to time constraints, we conduct supplementary experiments under Protocols 1 and 2. The results (reported as F1 / AUC) are summarized in the table below:
>
> | **Method** | **Train\Test** | **DocT-S** | **FSTS-S** | **FSTS-2k** | **AFAC** | **CertificatePS** | **STFD** | **FindIt** | **Average** |
> |------------|----------------|------------|-------------|--------------|---------------|------------------|----------|------------|-------------|
> | **TruFor** | DocT-T   | .516 / .982 | .400 / .901 | .211 / .708 | .185 / .811 | .289 / .811 | .091 / .787 | .214 / .811 | .198 / .785 |
> |            | FSTS-T | .270 / .868 | .775 / .980 | .683 / .952 | .638 / .984 | .487 / .892 | .190 / .865 | .386 / .868 | .477 / .912 |
> |            | **Gain Δ** | −.247 / −.114 | .374 / .079 | .471 / .244 | .453 / .174 | .198 / .082 | .099 / .078 | .172 / .057 | .279 / .127 |
> | **HiFi-Net** | DocT-T   | .019 / .568 | .096 / .475 | .092 / .476 | .033 / .483 | .182 / .492 | .053 / .499 | .136 / .500 | .099 / .490 |
> |            | FSTS-T | .017 / .465 | .104 / .495 | .095 / .448 | .038 / .531 | .180 / .442 | .055 / .500 | .118 / .430 | .097 / .470 |
> |            | **Gain Δ** | −.002 / −.104 | .008 / .019 | .003 / −.028 | .006 / .048 | −.001 / −.050 | .002 / .001 | −.017 / −.069 | −.002 / −.020 |
> | **SparseViT** | DocT-T   | .385 / .970 | .157 / .881 | .177 / .718 | .024 / .881 | .242 / .807 | .058 / .802 | .159 / .793 | .132 / .800 |
> |            | FSTS-T | .177 / .879 | .508 / .951 | .565 / .946 | .381 / .957 | .292 / .831 | .125 / .812 | .213 / .819 | .315 / .873 |
> |            | **Gain Δ** | −.207 / −.091 | .351 / .070 | .388 / .228 | .356 / .076 | .050 / .024 | .068 / .010 | .054 / .026 | .183 / .073 |
> | **Mesorch** | DocT-T   | .360 / .969 | .335 / .919 | .250 / .769 | .108 / .898 | .332 / .847 | .050 / .792 | .254 / .845 | .199 / .830 |
> |            | FSTS-T | .225 / .850 | .764 / .986 | .638 / .966 | .581 / .983 | .388 / .875 | .112 / .852 | .348 / .875 | .413 / .910 |
> |            | **Gain Δ** | −.134 / −.119 | .429 / .066 | .388 / .197 | .473 / .085 | .056 / .028 | .062 / .060 | .094 / .030 | .215 / .080 |
>
> Then, we analyze the results as follows:
>
> - Under Protocol 1, the results are consistent with the conclusion reported in the main paper: models trained on synthetic data (Doc-T, FSTS-T) and tested on the corresponding synthetic test sets (Doc-S, FSTS-S) exhibit superior performance, validating the effectiveness of synthetic data training. Results for TruFor, HiFi-Net, SparseViT, and Mesorch follow similar trends.
>
> - Under Protocol 2, where models trained on synthetic data are tested on real-world datasets, HiFi-Net performs poorly regardless of the training source, indicating its limited adaptability to the T-IFL task. In contrast, **other SOTA methods consistently benefit from FSTS-generated training data, achieving substantial performance improvements. For example, the other three models (TruFor, SparseViT, and Mesorch) trained on FSTS-T outperform those trained on DocT-T with an average F1 gain exceeding 18%, and TruFor achieves an average F1 improvement of 27.9%.**
>
> These results further demonstrate the broad application potential and effectiveness of our FSTS framework across diverse model types and protocols.

---

> > ### Comment · Reviewer_1iDP · 2025-08-08
> > **Reviewer 1iDP**
> >
> > First I want to apologize the authors and ACs for the late response due to some personal issues. I have some common upon to the authors's rebuttal
> >
> > 1. It is persuasive with the author's rebuttal for the question (1) and (2). All my concerns in these questions are addressed well.  However, in the IFDL task, one image should be classified to the edited or authentic one before localizing the edited region. Therefore, the style transfer is still need to be considered. Nevertheless, I agree with the authors that these kind of tasks are rarely observed, but it is worth considering.
> >
> > 2. The comparative results with TruFor and HiFi seems promising, demonstrating the effectiveness of the dataset and I hope that it would be published after the final decision. I am happy to raise my score.

---

> ### Author Response · Authors · 2025-08-06
>
> Thank you again for your thoughtful review.
> If you have any further questions or would like to see additional experiments, please let us know as soon as possible so that we have adequate time to prepare and share the results before the discussion period ends.

---

### Author Response · Authors · 2025-08-05

## Summary of Reviewers’ Feedback

We thank all reviewers for their constructive and insightful feedback. Across the reviews, several key strengths of our work were consistently recognized:

- **Clear motivation addressing a real gap**
  Reviewers agreed that our work is well‑motivated and addresses a meaningful gap in T‑IFL research. Reviewer k4rU noted that *“the motivation is clear and the method is straightforward but effective”*, while Reviewer aQSz described it as *“well‑motivated and thoroughly evaluated, showing clear benefits for model generalization”*. Reviewer asrA further emphasized that the paper *“addresses a core limitation in T‑IFL by shifting focus from visible dataset attributes to the invisible distribution of tampering behaviors”*.

- **Structured and interpretable framework**
  The proposed approach was praised for its structured and interpretable design. Reviewer k4rU found that *“introducing spectral analysis into structured and interpretable data analysis is very interesting”*. Reviewer aQSz described our method as *“a hierarchical Fourier series‑inspired framework to model real‑world tampering parameters; solid and of interest to the forensic community”*, and Reviewer asrA called it *“a novel and interpretable synthesis framework using a basis‑and‑weight formulation”*.

- **Comprehensive evaluation and consistent gains**
  The thoroughness of our experiments and the consistency of improvements were noted by multiple reviewers. Reviewer aQSz observed that *“models trained on FSTS consistently outperform those trained on other synthetic datasets”*, while Reviewer asrA remarked on the *“significantly improved generalization to real‑world datasets”*. Reviewer 1iDP highlighted the robustness of our results, pointing to *“consistent gains across four protocols”*.

- **Clear presentation**
  Reviewers commended the clarity and logical structure of the presentation. Reviewer k4rU commented that *“charts are clear and logically organized”*, Reviewer aQSz noted that *“the pipeline is well‑motivated and logically presented”*, and Reviewer asrA summarized the writing as *“well‑written, clearly presented, timely contribution”*.


---

## How We Addressed the Main Concerns

### 1. Methodology & Experiments
- **Clarification of tampering process and representative types**
  (1) For Reviewer asrA: We added explicit step‑by‑step descriptions for all five tampering types in Section B.2 of the supplementary material, complementing Figure 1(II) to clearly illustrate their implementation details for readers from other domains.
  (2) For Reviewer 1iDP: We clarified the scope of our task: while style‑transfer and semantic manipulations are technically possible, they are rarely observed in practical T‑IFL. Our work therefore focuses on more prevalent and relevant manipulation types.

- **Inclusion of additional SOTA baselines and results**
  (1) For Reviewer 1iDP: We clarified our baseline selection strategy and added experiments with TruFor, HiFi‑Net, and two recent methods (SparseViT and Mesorch). FSTS‑trained models consistently outperform DocT‑trained models across multiple protocols.

- **Computational requirements and low‑resource accessibility**
  (1) For Reviewer 1iDP: We clarified the computational resource requirements for FSTS‑based training, providing details on GPU memory usage, training time, and configurations. These confirm that FSTS can be trained on widely available GPUs without modification.

- **Inclusion of related work and broader context**
  (1) For Reviewer k4rU: We will expand the related work with frequency analysis methods, add the recommended citations, and clarify our spectral analysis approach in the broader forensic context.

### 2. Dataset & Resources
- **Dataset accessibility and metadata completeness**
  (1) For Reviewer k4rU: In response to your concern, We re‑checked dataset availability, clarified metadata limitations, and will provide standardized metadata in the final release.

- **Future resource release**
  (1) For Reviewers aQSz, asrA & k4rU: In response to your interest in additional synthesis‑related resources, we will release generation scripts, illustrative video samples, and representative failure examples in the revised version (subject to policy). These could not be included during the rebuttal phase but will be provided to support reproducibility and understanding.

### 3. Presentation & Formatting
- **Minor presentation issues**
  (1) For Reviewers asrA: We corrected the abstract typo and will proofread the final version for consistency.

  (2) For Reviewers k4rU: We will ensure consistent citation formatting for conferences and journals, unify reference styles, and standardize capitalization.

---

**We thank the reviewers for their valuable comments. The rebuttal phase has clarified our contributions and addressed concerns. We welcome any further discussion during the review period.**

---

### Decision · Program_Chairs · 2025-09-18

**Decision:**

Accept (poster)

**Comment:**

This paper proposes a Fourier series-inspired hierarchical modeling framework for real-world tampering parameters, enabling the principled synthesis of text-image forgeries grounded in empirical manipulation patterns. The contribution is both technically robust and of notable relevance to forensic inquiry. Some lacks of state-of-the-art comparisons which were raised during rebuttal phase, were properly resolved in rebuttal phase. Further, indications' mistakes and omitting the previous works were also responded.

This paper is **recommended for acceptance** with the following reasons: (1) the paper fully addressed concerns from four peer reviewers and then all reviewers were agreed to accept the manuscript, (2) the proposed method contains practical merits toward real-world text forgeries by providing a scalable Fourier analytic model, which make sounds and solid statements.